# Mixture Proportion Estimation and PU Learning: A Modern Approach

**Saurabh Garg[1], Yifan Wu[1], Alex Smola[2], Sivaraman Balakrishnan[1], Zachary C. Lipton[1]**
[1]Carnegie Mellon University
[2]Amazon Web Services

## Abstract

Given only positive examples and unlabeled examples (from both positive and negative classes), we might hope nevertheless to estimate an accurate positive-versus-negative classifier. Formally, this task is broken down into two subtasks: (i) *Mixture Proportion Estimation* (MPE)—determining the fraction of positive examples in the unlabeled data; and (ii) *PU-learning*—given such an estimate, learning the desired positive-versus-negative classifier. Unfortunately, classical methods for both problems break down in high-dimensional settings. Meanwhile, recently proposed heuristics lack theoretical coherence and depend precariously on hyperparameter tuning. In this paper, we propose two simple techniques: *Best Bin Estimation* (BBE) (for MPE); and *Conditional Value Ignoring Risk* (CVIR), a simple objective for PU-learning. Both methods dominate previous approaches empirically, and for BBE, we establish formal guarantees that hold whenever we can train a model to cleanly separate out a small subset of positive examples. Our final algorithm $(\text{TED})^n$, alternates between the two procedures, significantly improving both our mixture proportion estimator and classifier[1].

## 1 Introduction

When deploying $k$-way classifiers in the wild, what can we do when confronted with data from a previously unseen class $(k + 1)$? Theory dictates that learning under distribution shift is impossible absent assumptions. And yet people appear to exhibit this capability routinely. Faced with new surprising symptoms, doctors can recognize the presence of a previously unseen ailment and attempt to estimate its prevalence. Similarly, naturalists can discover new species, estimate their range and population, and recognize them reliably going forward.

To begin making this problem tractable, we might make the label shift assumption [37, 41, 29], i.e., that while the class balance $p(y)$ can change, the class conditional distributions $p(x|y)$ do not. Moreover, we might begin by focusing on the base case, where only one class has been seen previously, i.e., $k = 1$. Here, we possess (labeled) positive data from the source distribution, and (unlabeled) data from the target distribution, consisting of both positive and negative instances. This problem has been studied in the literature as *learning from positive and unlabeled data* [8, 27] and has typically been broken down into two subtasks: (i) Mixture Proportion Estimation (MPE) where we estimate $\alpha$, the fraction of positives among the unlabeled examples; and (ii) PU-learning where this estimate is incorporated into a scheme for learning a Positive-versus-Negative (PvN) binary classifier.

Traditionally, MPE and PU-learning have been motivated by settings involving large databases where unlabeled examples are abundant and a small fraction of the total positives have been extracted. For example, medical records might be annotated indicating the presence of certain diagnoses, but the unmarked passages are not necessarily negative. This setup has also been motivated by protein and

---

[1]Code is available at https://github.com/acmi-lab/PU_learning

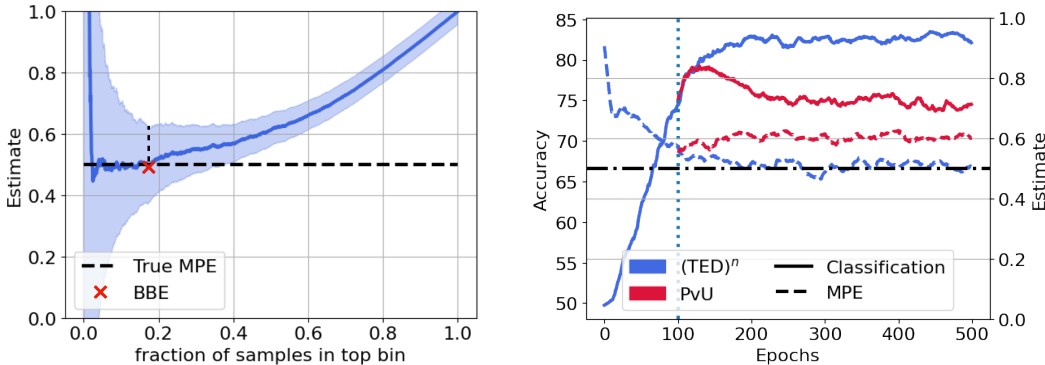

Figure 1: *Illustration of proposed methods.* **(left)** Estimate of $\alpha$ with varying fraction of unlabeled examples in the top bin. The shaded region highlights the upper and lower confidence bounds. BBE selects the top bin that minimizes the upper confidence bound. **(right)** Accuracy and MPE estimate as training proceeds. Till 100-th epoch (vertical line), we perform PvU training, i.e., warm start for $(\text{TED})^n$. Post 100-th epoch, we continue with both $(\text{TED})^n$ and PvU training. Note that $(\text{TED})^n$ improves both classification accuracy and MPE compared to PvU training. Results with Resnet-18 on binary-CIFAR. For details and comparisons with other methods, see Sec. 6.

gene identification [16]. Databases in molecular biology often contain lists of molecules known to exhibit some characteristic of interest. However, many other molecules may exhibit the desired characteristic, even if this remains unknown to science.

Many methods have been proposed for both MPE [16, 12, 39, 35, 21, 4, 36, 20] and PU-learning [14, 11, 23]. However, classical MPE methods break down in high-dimensional settings [35] or yield estimators whose accuracy depends on restrictive conditions [12, 39]. On the other hand, most recent proposals either lack theoretical coherence, rely on heroic assumptions, or depend precariously on tuning hyperparameters that are, by the very problem setting, untunable. For PU learning, Elkan and Noto [16] suggest training a classifier to distinguish positive from unlabeled data followed by a rescaling procedure. Du Plessis et al. [11] suggest an unbiased risk estimation framework for PU learning. However, these methods fail badly when applied with model classes capable of overfitting and thus implementations on high-dimensional datasets rely on extensive hyperparameter tuning and additional ad-hoc heuristics that do not transport effectively across datasets.

In this paper, we propose (i) Best Bin Estimation (BBE), an effective technique for MPE that produces consistent estimates $\widehat{\alpha}$ under mild assumptions and admits finite-sample statistical guarantees achieving the desired $O(1/\sqrt{n})$ rates; and (ii) learning with the Conditional Value Ignoring Risk (CVIR) objective, which discards the highest loss $\widehat{\alpha}$ fraction of examples on each training epoch, removing the incentive to overfit to the unlabeled positive examples. Both methods are simple to implement, compatible with arbitrary hypothesis classes (including deep networks), and dominate existing methods in our experimental evaluation. Finally, we combine the two in an iterated Transform-Estimate-Discard $(\text{TED})^n$ framework that significantly improves both MPE estimation error and classifier error.

We build on label shift methods [29, 3, 2, 34, 17], that leverage black-box classifiers to reduce dimensionality, estimating the target label distribution as a functional of source and target push-forward distributions. While label shift methods rely on classifiers trained to separate previously seen classes, BBE is able to exploit a Positive-versus-Unlabeled (PvU) target classifier, which gives each input a score indicating how likely it is to be a positive sample. In particular, BBE identifies a threshold such that by estimating the ratio between the fractions of positive and unlabeled points receiving scores above the threshold, we obtain the mixture proportion $\alpha$.

BBE works because in practice, for many datasets, PvU classifiers, even when uncalibrated, produce outputs with near monotonic calibration diagrams. Higher scores correspond to a higher proportion of positives, and when the positive data contains a separable sub-domain, i.e., a region of the input space where only the positive distribution has support, classifiers often exhibit a threshold above which the *top bin* contains mostly positive examples. We show that the existence of a (nearly) pure top bin is sufficient for BBE to produce a (nearly) consistent estimate $\widehat{\alpha}$, whose finite sample convergence

rates depend on the fraction of examples in the bin and whose bias depends on the *purity* of the bin. Crucially, we can estimate the optimal threshold from data.

We conduct a battery of experiments both to empirically validate our claim that BBE's assumptions are mild and frequently hold in practice, and to establish the outperformance of BBE, CVIR, and $(\text{TED})^n$ over the previous state of the art. We first motivate BBE by demonstrating that in practice PvU classifiers tend to isolate a reasonably large, reasonably pure top bin. We then conduct extensive experiments on semi-synthetic data, adapting a variety of binary classification datasets to the PU learning setup and demonstrating the superior performance of BBE and PU-learning with the CVIR objective. Moreover, we show that $(\text{TED})^n$, which combines the two in an iterative fashion, improves significantly over previous methods across several architectures on a range of image and text datasets.

## 2 Related Work

Research on MPE and PU learning date to [9, 8, 27] (see review by [5]). Elkan and Noto [16] first proposed to leverage a PvU classifier to estimate the mixture proportion. Du Plessis and Sugiyama [13] propose a different method for estimating the mixture coefficient based on Pearson divergence minimization. While they do not require a PvU classifier, they suffer the same shortcoming. Both methods require that the positive and negative examples have disjoint support. Our requirements are considerably milder. Blanchard et al. [6] observe that without assumptions on the underlying positive and negative distributions, the mixture proportion is not identifiable. Furthermore, [6] provide an *irreducibility* condition that identifies $\alpha$ and propose an estimator that converges to the true $\alpha$. While their estimator can converge arbitrarily slowly, Scott [39] showed faster convergence ($\mathcal{O}(1/\sqrt{n})$) under stronger conditions. Unfortunately, despite its appealing theoretical properties Blanchard et al. [6]'s estimator is computationally infeasible. Building on Blanchard et al. [6], Sanderson and Scott [38] and Scott [39] proposed estimating the mixture proportion from a ROC curve constructed for the PvU classifier. However, when the PvU classifier is not perfect, these methods are not clearly understood. Ramaswamy et al. [35] proposed the first computationally feasible algorithm for MPE with convergence guarantees to the true proportion. Their method KM, requires embedding distributions onto an RKHS. However, their estimator underperforms on high dimensional datasets and scales poorly with large datasets. Bekker and Davis [4] proposed TIcE, hoping to identify a positive subdomain in the input space using decision tree induction. This method also underperforms in high-dimensional settings.

In the most similar works, Jain et al. [21] and Ivanov [20] explore dimensionality reduction using a PvU classifier. Both methods estimate $\alpha$ through a procedure operating on the PvU classifier's output. However, neither methods has provided theoretical backing. [20] concede that their method often fails and returns a zero estimate, requiring that they fall back to a different estimator. Moreover while both papers state that their method require the Bayes-optimal PvU classifier to identify $\alpha$ in the transformed space, we prove that even when hypothesis class is well specified for PvN learning, PvU training can fail to recover the Bayes-optimal scoring function. Furthermore, we also show that the heuristic estimator in Scott [39] can be thought of as using PvU classifier for dimensionality reduction. While this heuristic is similar to our estimator in spirit, we show that the functional form of their estimator is different from ours and note that their heuristic enjoys no theoretical guarantee. By contrast, our estimator BBE is theoretically coherent under mild conditions and outperforms all of these methods empirically.

Given $\alpha$, Elkan and Noto [16] propose a transformation via Bayes rule to obtain the PvN classifier. They also propose a weighted objective, with weights given by the PvU classifier. Other propose unbiased risk estimators [14, 11] which require the mixture proportion $\alpha$. Du Plessis et al. [14] proposed an unbiased estimator with non-convex loss functions satisfying a specific symmetric condition, and subsequently Du Plessis et al. [11] generalized it to convex loss functions (denoted uPU in our experiments). in our experiments. Noting the problem of overfitting in modern overparameterized models, Kiryo et al. [23] propose a regularized extension that clips the loss on unlabeled data to zero. This is considered the current state-of-the-art in PU literature (denoted nnPU in our experiments). More recently, Ivanov [20] proposed DEDPUL, which finetunes the PvU classifiers using several heuristics, Bayes rule, and Expectation Maximization (EM). Since their method only applies a post-processing procedure, they rely on a good domain discriminator classifier in the first place and several hyperparameters for their heuristics. Several classical methods attempt to learn weights that identify reliable negative examples [30, 28, 26, 31, 44]. However, these earlier methods have not been successful with modern deep learning models.

---

**Algorithm 1** Best Bin Estimation (BBE)

---

**input** : Validation positive ($X_p$) and unlabeled ($X_u$) samples. Blackbox model classifier $\widehat{f} : \mathcal{X} \to$ [0, 1]. Hyperparameter $0 < \delta, \gamma < 1$.

1: $Z_p, Z_u = f(X_p), f(X_u)$.

2: $\widehat{q}_u(z), \widehat{q}_p(z) = \frac{\sum_{z_i \in Z_p} \mathbb{I}[z_i \geqslant z]}{n_p}, \frac{\sum_{z_i \in Z_u} \mathbb{I}[z_i \geqslant z]}{n_u}$ for all $z \in [0, 1]$.

3: Estimate $\widehat{c} := \arg\min_{c \in [0,1]} \left( \frac{\widehat{q}_u(c)}{\widehat{q}_p(c)} + \frac{1+\gamma}{\widehat{q}_p(c)} \left( \sqrt{\frac{\log(4/\delta)}{2n_u}} + \sqrt{\frac{\log(4/\delta)}{2n_p}} \right) \right).$

**output** : $\widehat{\alpha} := \frac{\widehat{q}_u(\widehat{c})}{\widehat{q}_p(\widehat{c})}$

---

## 3 Problem Setup

By $\|\cdot\|$ and $\langle \cdot, \cdot \rangle$, we denote the Euclidean norm and inner product, respectively. For a vector $v \in \mathbb{R}^d$, we use $v_j$ to denote its $j^{\text{th}}$ entry, and for an event $E$, we let $\mathbb{I}[E]$ denote the binary indicator of the event. By $|A|$, we denote the cardinality of set $A$. Let $\mathcal{X} \in \mathbb{R}^d$ be the input space and $\mathcal{Y} = \{-1, +1\}$ be the output space. Let $\mathrm{P} : \mathcal{X} \times \mathcal{Y} \to [0, 1]$ be the underlying joint distribution and let $p$ denote its corresponding density.

Let $\mathrm{P}_p$ and $\mathrm{P}_n$ be the class-conditional distributions for positive and negative class and $p_p(x) = p(x|y = +1)$ and $p_n(x) = p(x|y = -1)$ be the corresponding class-conditional densities. $\mathrm{P}_u$ denotes the distribution of the unlabeled data and $p_u$ denotes its density. Let $\alpha \in [0, 1]$ be the fraction of positives among the unlabeled population, i.e., $\mathrm{P}_u = \alpha \mathrm{P}_p + (1 - \alpha)\mathrm{P}_n$. When learning from positive and unlabeled data, we obtain i.i.d. samples from the positive (class-conditional) distribution, which we denote as $X_p = \{x_1, x_2, \ldots, x_{n_p}\} \sim \mathrm{P}_p^{n_p}$ and i.i.d samples from unlabeled distribution as $X_u = \{x_{n_p+1}, x_{n_p+2}, \ldots, x_{n_p+n_u}\} \sim \mathrm{P}_u^{n_u}$.

MPE is the problem of estimating $\alpha$. Absent assumptions on $\mathrm{P}_p$, $\mathrm{P}_n$ and $\mathrm{P}_u$, the mixture proportion $\alpha$ is not identifiable [6]. Indeed, if $\mathrm{P}_u = \alpha \mathrm{P}_p + (1 - \alpha)\mathrm{P}_n$, then any alternate decomposition of the form $\mathrm{P}_u = (\alpha - \gamma)\mathrm{P}_p + (1 - \alpha + \gamma)\mathrm{P}'_n$, for $\gamma \in [0, \alpha)$ and $\mathrm{P}'_n = (1 - \alpha + \gamma)^{-1}(\gamma \mathrm{P}_p + (1 - \alpha)\mathrm{P}_n)$, is also valid. Since we do not observe samples from the distribution $\mathrm{P}_n$, the parameter $\alpha$ is not identifiable. Blanchard et al. [6] formulate an *irreducibility* condition under which $\alpha$ is identifiable. Intuitively, the condition restricts $\mathrm{P}_n$ to ensure that it can not be a (non-trivial) mixture of $\mathrm{P}_p$ and any other distribution. While this irreducibility condition makes $\alpha$ identifiable, in the worst-case, the parameter $\alpha$ can be difficult to estimate and any estimator must suffer an arbitrarily slow rate of convergence [6]. In this paper, we propose mild conditions on the PvU classifier that make $\alpha$ identifiable and allows us to derive finite-sample convergence guarantees.

With PU learning, the aim is to learn a classifier $f : \mathcal{X} \to [0, 1]$ to approximate $p(y = +1|x)$. We assume that we are given a loss function $\ell : [0, 1] \times \mathcal{Y} \to \mathbb{R}$, such that $\ell(z, y)$ is the loss incurred by predicting $z$ when the true label is $y$. For a classifier $f$ and a sampled set $X = \{x_1, x_2, \ldots, x_n\}$, we let $\widehat{L}^+(f; X) = \sum_{i=1}^n \ell(f(x_i), +1)/n$ denote the loss when predicting the samples as positive and $\widehat{L}^-(f; X) = \sum_{i=1}^n \ell(f(x_i), -1)/n$ the loss when predicting the samples as negative. For a sample set $X$ each with true label $y$, we define 0-1 error as $\widehat{\mathcal{E}}^y(f; X) = \sum_{i=1}^n \mathbb{I}[y(f(x_i) - t) \leqslant 0]/n$ for some predefined threshold $t$. Unless stated otherwise, the threshold is assumed to be 0.5.

## 4 Mixture Proportion Estimation

In this section, we introduce BBE, a new method that leverages a blackbox classifier $f$ to perform MPE and establish convergence guarantees. All proofs are relegated to App. B. To begin, we assume access to a fixed classifier $f$. For intuition, you may think of $f$ as a PvU classifer trained on some portion fo the positive and unlabeled examples. In Sec. 5, we discuss other ways to obtain a suitable classifier from PU data.

We now introduce some additional notation. Assume $f$ transforms an input $x \in \mathcal{X}$ to $z \in [0, 1]$, i.e., $z = f(x)$. For given probability density function $p$ and a classifier $f$, define a function $q(z) = \int_{A_z} p(x) dx$, where $A_z = \{x \in \mathcal{X} : f(x) \geqslant z\}$ for all $z \in [0, 1]$. Intuitively, $q(z)$ captures the cumulative density of points in a top bin, the proportion of input domain that is assigned a value larger than $z$ by the classifier $f$ in the transformed space. We now define an empirical estimator $\widehat{q}(z)$ given a

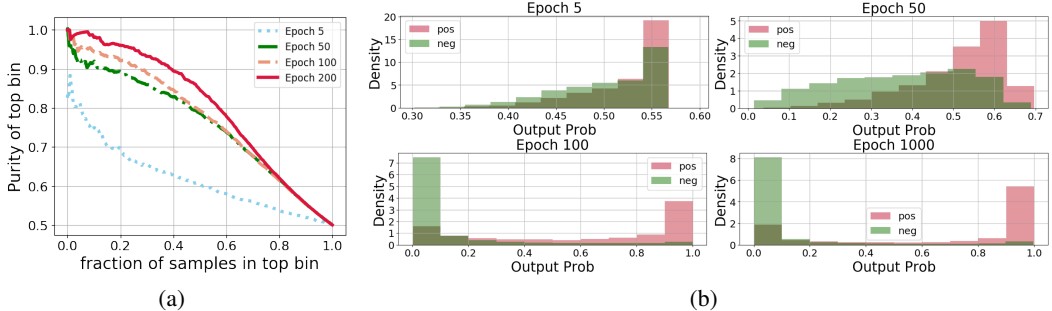

(a)                                                                                          (b)

Figure 2: (a) Purity and size (in terms of fraction of unlabeled samples) in the top bin and (b) Distribution of predicted probabilities (of being positive) for unlabeled training data as training proceeds with $(TED)^n$. Results with ResNet-18 on binary-CIFAR. As in Fig. 1, we fix $W$ at 100. In App. G.4, we show that as PvU training proceeds, the purity of top bin degrades and the distribution of predicted probabilities of positives and negatives become less and less separable.

set $X = \{x_1, x_2, \dots, x_n\}$ sampled iid from $p(x)$. Let $Z = f(X)$. Define $\widehat{q}(z) = \sum_{i=1}^n \mathbb{I}[z_i \geqslant z]/n$. For each pdf $p_p$, $p_n$ and $p_u$, we define $q_p$, $q_n$ and $q_u$ respectively.

Without any assumptions on the underlying distribution and the classifier $f$, we aim to estimate $\alpha^* = \min_{c \in [0,1]} q_u(c)/q_p(c)$ with BBE. Later, under one mild assumption that empirically holds across numerous PU datasets, we show that $\alpha^* = \alpha$, i.e., $\alpha^*$ matches the true mixture proportion $\alpha$.

Our procedure proceeds as follows: First, given a held-out dataset of positive $(X_p)$ and unlabeled examples $(X_u)$, we push all examples through the classifier $f$ to obtain one-dimensional outputs $Z_p = f(X_p)$ and $Z_u = f(X_u)$. Next, with $Z_p$ and $Z_u$, we estimate $\widehat{q}_p$ and $\widehat{q}_u$. Finally, we return the ratio $\widehat{q}_u(\widehat{c})/\widehat{q}_p(\widehat{c})$ at $\widehat{c}$ that minimizes the upper confidence bound (calculated using Lemma 1) at a pre-specified level $\delta$ and a fixed parameter $\gamma \in (0, 1)$. Our method is summarized in Algorithm 1. For theoretical guarantees, we multiply the confidence bound term with $1 + \gamma$ for a small positive constant $\gamma$. Refer to App. B.1 for details. We now show that the proposed estimator comes with the following guarantee:

**Theorem 1.** *Define* $c^* = \arg\min_{c \in [0,1]} q_u(c)/q_p(c)$. *For* $\min(n_p, n_u) \geqslant \frac{2\log(4/\delta)}{q_p(c^*)}$ *and for every* $\delta > 0$, *the mixture proportion estimator* $\widehat{\alpha}$ *defined in Algorithm 1 satisfies with probability* $1 - \delta$:

$$|\widehat{\alpha} - \alpha^*| \leqslant \frac{c}{q_p(c^*)} \left( \sqrt{\frac{\log(4/\delta)}{n_u}} + \sqrt{\frac{\log(4/\delta)}{n_p}} \right),$$

*for some constant* $c \geqslant 0$.

Theorem 1 shows that with high probability, our estimate is close to $\alpha^*$. The proof of the theorem is based on the following confidence bound inequality:

**Lemma 1.** *For every* $\delta > 0$, *with probability at least* $1 - \delta$, *we have for all* $c \in [0, 1]$

$$\left| \frac{\widehat{q}_u(c)}{\widehat{q}_p(c)} - \frac{q_u(c)}{q_p(c)} \right| \leqslant \frac{1}{\widehat{q}_p(c)} \left( \sqrt{\frac{\log(4/\delta)}{2n_u}} + \frac{q_u(c)}{q_p(c)} \sqrt{\frac{\log(4/\delta)}{2n_p}} \right).$$

Now, we discuss the convergence of our estimator to the true mixture proportion $\alpha$. Since, $p_u(x) = \alpha p_p(x) + (1 - \alpha)p_n(x)$, for all $x \in \mathcal{X}$, we have $q_u(z) = \alpha q_p(z) + (1 - \alpha)q_n(z)$, for all $z \in [0, 1]$.

**Corollary 1.** *Define* $c^* = \arg\min_{c \in [0,1]} q_n(c)/q_p(c)$. *Assume* $\min(n_p, n_u) \geqslant \frac{2\log(4/\delta)}{q_p(c^*)}$. *For every* $\delta > 0$, $\widehat{\alpha}$ *(in Algorithm 1) satisfies with probability* $1 - \delta$:

$$\alpha - \frac{c_1}{q_p(c^*)} \left( \sqrt{\frac{\log(4/\delta)}{n_u}} + \sqrt{\frac{\log(4/\delta)}{n_p}} \right) \leqslant \widehat{\alpha}, \text{ and}$$

$$\widehat{\alpha} \leqslant \alpha + (1 - \alpha)\frac{q_n(c^*)}{q_p(c^*)} + \frac{c_2}{q_p(c^*)} \left( \sqrt{\frac{\log(4/\delta)}{n_u}} + \sqrt{\frac{\log(4/\delta)}{n_p}} \right),$$

*for some constant* $c_1, c_2 \geqslant 0$.

---

**Algorithm 2** PU learning with Conditional Value Ignoring Risk (CVIR) objective

---

**input** : Labeled positive training data $(X_p)$ and unlabeled training samples $(X_u)$. Mixture proportion estimate $\alpha$.

  1: Initialize a training model $f_\theta$ and an stochastic optimization algorithm $\mathcal{A}$.
  2: $X_n := X_u$.
  3: **while** training error $\widehat{\mathcal{E}}^+(f_\theta; X_p) + \widehat{\mathcal{E}}^-(f_\theta; X_n)$ is not converged **do**
  4:     Rank samples $x_u \in X_u$ according to their loss values $\ell(f_\theta(x_u), -1)$.
  5:     $X_n := X_{u,1-\alpha}$ where $X_{u,1-\alpha}$ denote the lowest ranked $1 - \alpha$ fraction of samples.
  6:     Shuffle $(X_p, X_n)$ into $B$ mini-batches. With $(X_p^i, X_n^i)$ we denote $i$-th mini-batch.
  7:     **for** $i = 1$ to $B$ **do**
  8:       Set the gradient $\nabla_\theta \left[ \alpha \cdot \widehat{L}^+(f_\theta; X_p^i) + (1 - \alpha) \cdot \widehat{L}^-(f_\theta; X_n^i) \right]$ and update $\theta$ with algo. $\mathcal{A}$.
  9:     **end for**
10: **end while**

**output** : Trained classifier $f_\theta$

---

As a corollary to Theorem 1, we show that our estimator $\widehat{\alpha}$ converges to the true $\alpha$ with convergence rate $\min(n_p, n_u)^{-1/2}$, as long as there exist a threshold $c_f \in (0, 1)$ such that $q_p(c_f) \geqslant \epsilon_p$ and $q_n(c_f) = 0$ for some constant $\epsilon_p > 0$. We refer to this condition as the *pure positive bin* property.

Note that in a more general case, our bound in Corollary 1 captures the tradeoff due to the proportion of negative examples in the top bin (bias) versus the proportion of positives in the top bin (variance).

**Empirical Validation** We now empirically validate the positive pure top bin property (Fig. 2). We observe that as PvU training proceeds, purity of the top bin improves for a fixed fraction of samples in the top bin. Moreover, this behavior becomes more pronounced when learning a PvU classifier with the CVIR objective proposed in the following section.

**Comparison with existing methods** Due to the intractability of Blanchard et al. [6] estimator, Scott [39] implements a heuristic based on identifying a point on the AUC curve such that the slope of the line segment between this point and (1,1) is minimized. While this approach is similar in spirit to our BBE method, there are some striking differences. First, the heuristic estimator in Scott [39] provides no theoretical guarantees, whereas we provide guarantees that BBE will converge to the best estimate achievable over all choices of the bin size and provide consistent estimates whenever a pure top bin exists. Second, while both estimates involve thresholds, the functional form of the estimates are different. Corroborating theoretical results of BBE, we observe that the choices in BBE create substantial differences in the empirical performance as observed in App. C. We work out details of comparison between Scott [39] heuristic and BBE in App. C.

On the other hand, recent works [21, 20] that use PvU classifier for dimensionality reduction, discuss Bayes optimality of the PvU classifier (or its one-to-one mapping) as a sufficient condition to preserve $\alpha$ in transformed space. By contrast, we show that the milder pure positive bin property is sufficient to guarantee consistency and achieve $\mathcal{O}(1/\sqrt{n})$ rates. Furthermore, in a simple toy setup in App. D, we show that even when the hypothesis class is well specified for PvN learning, it will not in general contain the Bayes optimal PvU classifier and thus PvU training will not recover the Bayes-optimal scoring function, even in population. Contrarily, we show that any monotonic mapping of the Bayes-optimal PvU scoring function induces a positive pure top bin property. We leave further theoretical investigations concerning conditions under which a pure positive top bin arises to future work.

## 5 PU-Learning

Given positive and unlabeled data, we hope not only to identify $\alpha$, but also to obtain a classifier that distinguishes effectively between positive and negative samples. In supervised learning with separable data (e.g., cleanly labeled image data), overparameterized models generalize well even after achieving near-zero training error. However, with PvU training over-parameterized models can memorize the unlabeled positives, assigning them confidently to the negative class, which can severely hurt generalization on PN data [43]. Moreover, while unbiased losses exist that estimate the PvN loss given PU data and the mixture proportion $\alpha$, this unbiasedness only holds before the loss is optimized, and becomes ineffective with powerful deep learning models capable of memorization.

---
**Algorithm 3** Transform-Estimate-Discard (TED)$^n$

---
**input** : Positive data ($X_p$) and unlabeled samples ($X_u$). Hyperparameter $W, \delta$.
1: Initialize a training model $f_\theta$ and an stochastic optimization algorithm $\mathcal{A}$.
2: Randomly split positive and unlabeled data into training $X_p^1, X_u^1$ and hold-out set ($X_p^2, X_u^2$).
3: $X_n^1 := X_u^1$.
  {// Warm start with domain discrimination training}
4: **for** $i = 1$ to $W$ **do**
5:   Shuffle $(X_p^1, X_n^1)$ into $B$ mini-batches. With $(X_p^{1i}, X_n^{1i})$ we denote $i$-th mini-batch.
6:   **for** $i = 1$ to $B$ **do**
7:     Set the gradient $\nabla_\theta \left[ \widehat{L}^+(f_\theta; X_p^{1i}) + \widehat{L}^-(f_\theta; X_n^{1i}) \right]$ and update $\theta$ with algorithm $\mathcal{A}$.
8:   **end for**
9: **end for**
10: **while** training error $\widehat{\mathcal{E}}^+(f_\theta; X_p^1) + \widehat{\mathcal{E}}^-(f_\theta; X_n^1)$ is not converged **do**
11:   Estimate $\widehat{\alpha}$ using Algorithm 1 with $(X_p^2, X_u^2)$ and $f_\theta$ as input.
12:   Rank samples $x_u \in X_u^1$ according to their loss values $l(f_\theta(x_u), -1)$.
13:   $X_n^1 := X_{u,1-\widehat{\alpha}}^1$ where $X_{u,1-\widehat{\alpha}}^1$ denote the lowest ranked $1 - \widehat{\alpha}$ fraction of samples.
14:   Train model $f_\theta$ for one epoch on $(X_p^1, X_n^1)$ as in Lines 4-7.
15: **end while**
**output** : Trained classifier $f_\theta$

---

A variety of heuristics, including ad-hoc early stopping criteria, have been explored [20], where training proceeds until the loss on unseen PU data ceases to decrease. However, this approach leads to severe under-fitting (results in App. G.2). On the other hand, by regularizing the loss function, nnPU Kiryo et al. [23] mitigates overfitting issues due to memorization.

However, we observe that nnPU still leaves a substantial accuracy gap when compared to a model trained just on the positive and negative (from the unlabeled) data (ref. experiment in App. G.1). This leads us to ask the following question: *can we improve performance over nnPU of a model just trained with PU data and bridge this gap?* In an ideal scenario, if we could identify and remove all the positive points from the unlabeled data during training then we can hope to achieve improved performance over nnPU. Indeed, in practice, we observe that in the initial stages of PvU training, the model assigns much higher scores to positives than to negatives in the unlabeled data (Fig. 2(b)).

Inspired by this observation, we propose CVIR, a simple yet effective objective for PU learning. Below, we present our method assuming an access to the true MPE. Later, we combine BBE with CVIR optimization, yielding (TED)$^n$, an alternating optimization that significantly improves both the BBE estimates and the PvU classifier.

Given a training set of positives $X_p$ and unlabeled $X_u$ and the mixture proportion $\alpha$, we begin by ranking the unlabeled data according the predicted probability (of being positive) by our classifier. Then, in every epoch of training, we create a (temporary) set of provisionally negative samples $X_n$ by removing $\alpha$ fraction of the unlabeled samples currently scored as most positive. Next, we update our classifier by minimize the loss on the positives $X_p$ and provisional negatives $X_n$ by treating them as negatives. We repeat this procedure until the training error on $X_p$ and $X_n$ converges. Likewise nnPU, note that this procedure does not need early stopping. Summary in Algorithm 2.

In App. E, we justify our loss function in the scenario when the positives and negatives are separable. For a more general scenario, we show that each step of our alternating procedure in CVIR cannot increase the population loss and hence, CVIR can only improve (or plateau) after every iteration.

**(TED)$^n$ Integrating BBE and CVIR**   We are now ready to present our algorithm Transfer, Estimate and Discard (TED)$^n$ that combines BBE and CVIR objective.

First, we observe the interaction between BBE and CVIR objective. If we have an accurate mixture proportion estimate, then it leads to improved classifier, in particular, we reject accurate number of prospective positive samples from unlabeled. Consequently, updating the classifier to minimize loss on positive versus retained unlabeled improves purity of top bin. This leads to an obvious alternating procedure where at each epoch, we first use BBE to estimate $\widehat{\alpha}$ and then update the classifier with CVIR objective with $\widehat{\alpha}$ as input. We repeat this until training error has not converged. Our method is summarized in Algorithm 3.

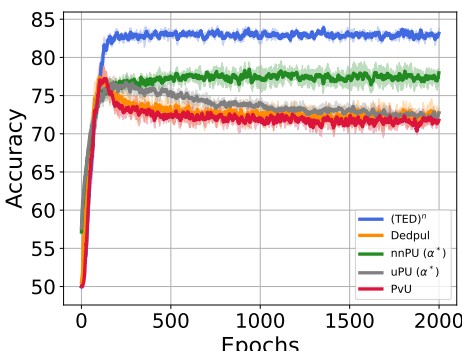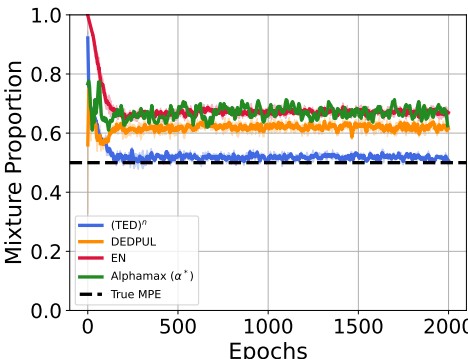

Figure 3: Epoch wise results with ResNet-18 trained on binary-CIFAR when $\alpha$ is 0.5. Parallel results on other datasets and architectures in App. G.3. For both classification and MPE, $(\text{TED})^n$ substantially improves over existing methods. Additionally, $(\text{TED})^n$ maintains the superior performance till convergence removing the need for early stopping. Results aggregated over 3 seeds.

Note that we need to warm start with PvU (positive versus negative) training, since in the initial stages mixture proportion estimate is often close to 1 rejecting all the unlabeled examples. However, in next section, we show that our procedure is not sensitive to the choice of number of warm start epochs and in a few cases with large datasets, we can even get away without warm start (i.e., $W = 0$) without hurting the performance. Moreover, recall that our aim is to distinguish positive versus negative examples among the unlabeled set where the proportion of positives is determined by the true mixture proportion $\alpha$. However, unlike CVIR, we do not re-weight the losses in $(\text{TED})^n$. While true MPE $\alpha$ is unknown, one natural choice is to use the estimate $\widehat{\alpha}$ at each iteration. However, in our initial experiments, we observed that re-weighted objective with estimate $\widehat{\alpha}$ led to comparatively poor classification performance due to presence of bias in estimate $\widehat{\alpha}$ in the initial iterations. We note that for deep neural networks (for which model mis-specification is seldom a prominent concern) and when the underlying classes are separable (as with most image datasets), it is known that importance weighting has little to no effect on the final classifier [7]. Therefore, we may not need importance-reweighting with $(\text{TED})^n$ on separable datasets. Consequently, following earlier works [23, 11] we do not re-weight the loss with our $(\text{TED})^n$ procedure. In future work, a simple empirical strategy can be explored where we first obtain an estimate of $\widehat{\alpha}$ by running the full $(\text{TED})^n$ procedure till convergence and then discarding the $(\text{TED})^n$ classifier, we use estimate $\widehat{\alpha}$ to train a fresh classifier with CVIR procedure.

Finally, we discuss an important distinction with Dedpul which is also an alternating procedure. While in our algorithm, after updating mixture proportion estimate we retrain the classifier, Dedpul fixes the classifier, obtains output probabilities and then iteratively updates the mixture proportion estimate (prior) and output probabilities (posterior). Dedpul doesn't re-train the classifier.

## 6 Experiments

Having presented our PU learning and MPE algorithms, we now compare their performance with other methods empirically. We mainly focus on vision and text datasets in our experiments. We include results on UCI datasets in App. G.7.

**Datasets and Evaluation**  We simulate PU tasks on CIFAR-10 [24], MNIST [25], and IMDb sentiment analysis [32] datasets. We consider binarized versions of CIFAR-10 and MNIST. On CIFAR-10 dataset, we consider two classification problems: (i) binarized CIFAR, i.e., first 5 classes vs rest; (ii) Dog vs Cat in CIFAR. Similarly, on MNIST, we consider: (i) binarized MNIST, i.e., digits 0-4 vs 5-9; (ii) MNIST17, i.e., digit 1 vs 7. IMDb dataset is binary. For MPE, we use a held out PU validation set. To evaluate PU classifiers, we calculate accuracy on held out positive versus negative dataset. For baselines that suffer from issues due to overfitting on unlabeled data, we report results with an *oracle early stopping* criterion. In particular, we report the accuracy averaged over 10 iterations of the best performing model as evaluated on positive versus negative data. Note that we use this oracle stopping criterion only for previously proposed methods and not for methods proposed

| Dataset | Model | $(TED)^n$ | BBE* | DEDPUL* | AlphaMax* | EN* | KM2 | TiCE |
|---|---|---|---|---|---|---|---|---|
| Binarized CIFAR | ResNet | **0.026** | 0.091 | 0.091 | 0.125 | 0.192 | | |
| | All Conv | 0.042 | **0.037** | 0.052 | 0.09 | 0.221 | 0.168 | 0.251 |
| | MLP | 0.225 | 0.177 | **0.138** | 0.3 | 0.372 | | |
| CIFAR Dog vs Cat | ResNet | **0.078** | 0.176 | 0.170 | 0.17 | 0.226 | 0.331 | 0.286 |
| | All Conv | **0.066** | 0.128 | 0.115 | 0.19 | 0.250 | | |
| Binarized MNIST | MLP | **0.024** | 0.032 | 0.031 | 0.090 | 0.080 | 0.029 | 0.056 |
| MNIST17 | MLP | **0.003** | 0.023 | 0.021 | 0.075 | 0.028 | 0.022 | 0.043 |
| IMDb | BERT | **0.008** | 0.011 | 0.016 | 0.07 | 0.12 | - | - |

Table 1: Absolute estimation error when $\alpha$ is 0.5. "*" denote oracle early stopping as defined in Sec. 6. $(TED)^n$ significantly reduces estimation error when compared with existing methods. Results reported by aggregating absolute error over 10 epochs and 3 seeds. For aggregate numbers with standard deviation see App. G.6.

| Dataset | Model | $(TED)^n$ (unknown $\alpha$) | CVIR (known $\alpha$) | PvU* (known $\alpha$) | DEDPUL* (unknown $\alpha$) | nnPU (known $\alpha$) | uPU* (known $\alpha$) |
|---|---|---|---|---|---|---|---|
| Binarized CIFAR | ResNet | **82.7** | 82.3 | 76.9 | 77.1 | 77.2 | 76.7 |
| | All Conv | 77.9 | **78.1** | 75.8 | 77.1 | 73.4 | 72.5 |
| | MLP | 64.2 | **66.9** | 61.6 | 62.6 | 63.1 | 64.0 |
| CIFAR Dog vs Cat | ResNet | **75.2** | 73.3 | 67.3 | 67.0 | 71.8 | 68.8 |
| | All Conv | **73.0** | 71.7 | 70.5 | 69.2 | 67.9 | 67.5 |
| Binarized MNIST | MLP | 95.6 | **96.3** | 94.2 | 94.8 | 96.1 | 95.2 |
| MNIST17 | MLP | **98.7** | **98.7** | 96.9 | 97.7 | 98.4 | 98.4 |
| IMDb | BERT | **87.6** | 87.4 | 86.1 | 87.3 | 86.2 | 85.9 |

Table 2: Accuracy for PvN classification with PU learning. "*" denote oracle early stopping as defined in Sec. 6. Results reported by aggregating over 10 epochs and 3 seeds. Both CVIR (with known MPE) and $(TED)^n$ (with unknown MPE) significantly improve over previous baselines with oracle early stopping and known MPE. For aggregate numbers with standard deviation see App. G.6.

in this work. This allows us to compare $(TED)^n$ with the best performance achievable by previous methods that suffer from over-fitting issues. With nnPU and $(TED)^n$, we report average accuracy over 10 iterations of the final model.

**Architectures** For CIFAR datasets, we consider (fully connected) multilayer perceptrons (MLPs) with ReLU activations, all convolution nets [40], and ResNet18 [19]. For MNIST, we consider multilayer perceptrons (MLPs) with ReLU activations For the IMDb dataset, we fine-tune an off-the-shelf uncased BERT model [10, 42]. We did not tune hyperparameters or the optimization algorithm—instead we use the same benchmarked hyperparameters and optimization algorithm for each dataset. For our method, we use cross-entropy loss. For uPU and nnPU, we use Adam [22] with sigmoid loss. We provide additional details about the datasets and architectures in App. F.

**Mixture Proportion Estimation** First, we discuss results for MPE (Table 1). We compare our method with KM2, TiCE, DEDPUL, AlphaMax and EN. Following earlier works [20, 35], we reduce datasets to 50 dimensions with PCA for KM2 and TiCE. We use existing implementation for other methods[2]. For BBE, DEDPUL and Alphamax, we use the same PvU classifier as input. On CIFAR datasets, convolutional classifier based estimators significantly outperform KM2 and TiCE. In contrast, the performance of KM2 is comparable to DEDPUL on MNIST datasets. On all datasets,

---

[2]DEDPUL: https://github.com/dimonenka/DEDPUL, KM: https://web.eecs.umich.edu/~cscott/code.html#kmpe, TiCE: https://dtai.cs.kuleuven.be/software/tice, and AlphaMax: https://github.com/Dzeiberg/AlphaMax

$(TED)^n$ achieves lowest estimation error. With the same blackbox classifier obtained with oracle early stopping, BBE performs similar or better than best alternate(s). Since overparamterized models start memorizing unlabeled samples negatives, the quality of MPE degrades substantially as PvU training proceeds for all methods but $(TED)^n$ as in Fig. 3 (epoch-wise results for on other tasks in App. G.3).

**Classification with known MPE**  Now, we discuss results for classification with known $\alpha$. We compare our method with uPU, nnPU[3], DEDPUL and PvU training. Although, we solve both MPE and classification, some comparison methods do not. Ergo, we compare our classification algorithm with known MPE (Algorithm 2).

To begin, first we note that nnPU and PvU training with CVIR doesn't need early stopping. For all other methods, we report the best performance dictated by the aforementioned oracle stopping criterion. On all datasets, PvU training with CVIR leads to improved classification performance when compared with alternate approaches (Table 2). Moreover, as training proceeds (Fig. 3), the performance of DEDPUL, PvU training and uPU substantially degrade. We repeated experiments with the early stopping criterion mentioned in DEDPUL (App. G.2), however, their early stopping criterion is too pessimistic resulting in poor results due to under-fitting.

**Classification with unknown MPE**  Finally, we evaluate $(TED)^n$, our alternating procedure for MPE and PU learning. Across many tasks, we observe substantial improvements over existing methods. Note that these improvements often are over an oracle early stopping baselines highlighting significance of our procedure.

In App. G.5, we show that our procedure is not sensitive to warm start epochs W, and in many tasks with $W = 0$, we observe minor-to-no differences in the performance of $(TED)^n$. While for the experiments in this section, we used fixed $W = 100$, in the Appendix we show behavior with varying W. We also include ablations with different mixture proportions $\alpha$.

## 7   Conclusion and Future Work

In this paper, we proposed two practical algorithms, BBE (for MPE) and CVIR optimization (for PU learning). Our methods outperform others empirically and BBE's mixture proportion estimates leverage black box classifiers to produce (nearly) consistent estimates with finite sample convergence guarantees whenever we possess a classifier with a (nearly) pure top bin. Moreover, $(TED)^n$ combines our procedures in an iterative fashion, achieving further gains. An important next direction is to extend our work to the multiclass problem [38], bridging work on label shift and PU learning. Here, we imagine that a deployed $k$-way classifier may encounter not only label shift among previously seen classes ([29, 17]) but also, potentially, instances from one previously unseen class. We also plan to investigate distributional properties under which we can hope to reliably or approximately satisfy the pure positive bin property with an off-the-shelf classifier trained on PvU data. While we improve significantly over previous PU methods, there is still a gap between $(TED)^n$'s performance and PvN training. We hope that our work can open a pathway towards further narrowing this gap.

## Acknowledgements

We thank anonymous reviewers for their feedback during NeurIPS 2021 review process. This material is based on research sponsored by Air Force Research Laboratory (AFRL) under agreement number FA8750-19-1-1000. The U.S. Government is authorized to reproduce and distribute reprints for Government purposes notwithstanding any copyright notation therein. The views and conclusions contained herein are those of the authors and should not be interpreted as necessarily representing the official policies or endorsements, either expressed or implied, of Air Force Laboratory, DARPA or the U.S. Government. SB acknowledges funding from the NSF grants DMS-1713003, DMS-2113684 and CIF-1763734, as well as Amazon AI and a Google Research Scholar Award. ZL acknowledges Amazon AI, Salesforce Research, Facebook, UPMC, Abridge, the PwC Center, the Block Center, the Center for Machine Learning and Health, and the CMU Software Engineering Institute (SEI) via Department of Defense contract FA8702-15-D-0002, for their generous support of ACMI Lab's research on machine learning under distribution shift.

---

[3]uPU and nnPU: https://github.com/kiryor/nnPUlearning

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
