# A  Appendix

# B  Proofs from Sec. 4

*Proof of Lemma 1.*  The proof primarily involves using DKW inequality [15] on $\widehat{q}_u(c)$ and $\widehat{q}_p(c)$ to show convergence to their respective means $q_u(c)$ and $q_p(c)$. First, we have

$$\left| \frac{\widehat{q}_u(c)}{\widehat{q}_p(c)} - \frac{q_u(c)}{q_p(c)} \right| = \frac{1}{\widehat{q}_u(c) \cdot q_u(c)} \left| \widehat{q}_u(c) \cdot q_p(c) - q_p(c) \cdot q_u(c) + q_p(c) \cdot q_u(c) - \widehat{q}_p(c) \cdot q_u(c) \right|$$

$$\leqslant \frac{1}{\widehat{q}_p(c)} \left| \widehat{q}_u(c) - q_u(c) \right| + \frac{q_u(c)}{\widehat{q}_p(c) \cdot q_u(c)} \left| \widehat{q}_p(c) - q_p(c) \right| . \tag{1}$$

Using DKW inequality, we have with probability $1 - \delta$: $|\widehat{q}_p(c) - q_p(c)| \leqslant \sqrt{\frac{\log(2/\delta)}{2n_p}}$ for all $c \in [0, 1]$. Similarly, we have with probability $1 - \delta$: $|\widehat{q}_u(c) - q_u(c)| \leqslant \sqrt{\frac{\log(2/\delta)}{2n_u}}$ for all $c \in [0, 1]$. Plugging this in (1), we have

$$\left| \frac{\widehat{q}_u(c)}{\widehat{q}_p(c)} - \frac{q_u(c)}{q_p(c)} \right| \leqslant \frac{1}{\widehat{q}_p(c)} \left( \sqrt{\frac{\log(4/\delta)}{2n_u}} + \frac{q_u(c)}{q_p(c)} \sqrt{\frac{\log(4/\delta)}{2n_p}} \right) .$$

$\square$

*Proof of Theorem 1.*  The main idea of the proof is to use the confidence bound derived in Lemma 1 at $\widehat{c}$ and use the fact that $\widehat{c}$ minimizes the upper confidence bound. The proof is split into two parts. First, we derive a lower bound on $\widehat{q}_p(\widehat{c})$ and next, we use the obtained lower bound to derive confidence bound on $\widehat{\alpha}$. All the statements in the proof simultaneously hold with probability $1 - \delta$. Recall,

$$\widehat{c} := \arg\min_{c \in [0,1]} \frac{\widehat{q}_u(c)}{\widehat{q}_p(c)} + \frac{1}{\widehat{q}_p(c)} \left( \sqrt{\frac{\log(4/\delta)}{2n_u}} + (1+\gamma)\sqrt{\frac{\log(4/\delta)}{2n_p}} \right) \qquad \text{and} \tag{2}$$

$$\widehat{\alpha} := \frac{\widehat{q}_u(\widehat{c})}{\widehat{q}_p(\widehat{c})} . \tag{3}$$

Moreover,

$$c^* := \arg\min_{c \in [0,1]} \frac{q_u(c)}{q_p(c)} \qquad \text{and} \qquad \alpha^* := \frac{q_u(c^*)}{q_p(c^*)} . \tag{4}$$

**Part 1:** We establish lower bound on $\widehat{q}_p(\widehat{c})$. Consider $c' \in [0, 1]$ such that $\widehat{q}_p(c') = \frac{\gamma}{2+\gamma}\widehat{q}_p(c^*)$. We will now show that Algorithm 1 will select $\widehat{c} < c'$. For any $c \in [0, 1]$, we have with with probability $1 - \delta$,

$$\widehat{q}_p(c) - \sqrt{\frac{\log(4/\delta)}{2n_p}} \leqslant q_p(c) \qquad \text{and} \qquad q_u(c) - \sqrt{\frac{\log(4/\delta)}{2n_u}} \leqslant \widehat{q}_u(c) . \tag{5}$$

Since $\frac{q_u(c^*)}{q_p(c^*)} \leqslant \frac{q_u(c)}{q_p(c)}$, we have

$$\widehat{q}_u(c) \geqslant q_p(c)\frac{q_u(c^*)}{q_p(c^*)} - \sqrt{\frac{\log(4/\delta)}{2n_u}} \geqslant \left( \widehat{q}_p(c) - \sqrt{\frac{\log(4/\delta)}{2n_p}} \right) \frac{q_u(c^*)}{q_p(c^*)} - \sqrt{\frac{\log(4/\delta)}{2n_u}} . \tag{6}$$

Therefore, at $c$ we have

$$\frac{\widehat{q}_u(c)}{\widehat{q}_p(c)} \geqslant \alpha^* - \frac{1}{\widehat{q}_p(c)} \left( \sqrt{\frac{\log(4/\delta)}{2n_u}} + \frac{q_u(c^*)}{q_p(c^*)} \sqrt{\frac{\log(4/\delta)}{2n_p}} \right) . \tag{7}$$

Using Lemma 1 at $c^*$, we have

$$\frac{\widehat{q}_u(c)}{\widehat{q}_p(c)} \geq \frac{\widehat{q}_u(c^*)}{\widehat{q}_p(c^*)} - \left(\frac{1}{\widehat{q}_p(c^*)} + \frac{1}{\widehat{q}_p(c)}\right)\left(\sqrt{\frac{\log(4/\delta)}{2n_u}} + \frac{q_u(c^*)}{q_p(c^*)}\sqrt{\frac{\log(4/\delta)}{2n_p}}\right) \tag{8}$$

$$\geq \frac{\widehat{q}_u(c^*)}{\widehat{q}_p(c^*)} - \left(\frac{1}{\widehat{q}_p(c^*)} + \frac{1}{\widehat{q}_p(c)}\right)\left(\sqrt{\frac{\log(4/\delta)}{2n_u}} + \sqrt{\frac{\log(4/\delta)}{2n_p}}\right), \tag{9}$$

where the last inequality follows from the fact that $\alpha^* = \frac{q_u(c^*)}{q_p(c^*)} \leq 1$. Furthermore, the upper confidence bound at $c$ is lower bound as follows:

$$\frac{\widehat{q}_u(c)}{\widehat{q}_p(c)} + \frac{1+\gamma}{\widehat{q}_p(c)}\left(\sqrt{\frac{\log(4/\delta)}{2n_u}} + \sqrt{\frac{\log(4/\delta)}{2n_p}}\right) \tag{10}$$

$$\geq \frac{\widehat{q}_u(c^*)}{\widehat{q}_p(c^*)} + \left(\frac{1+\gamma}{\widehat{q}_p(c)} - \frac{1}{\widehat{q}_p(c^*)} - \frac{1}{\widehat{q}_p(c)}\right)\left(\sqrt{\frac{\log(4/\delta)}{2n_u}} + \sqrt{\frac{\log(4/\delta)}{2n_p}}\right) \tag{11}$$

$$= \frac{\widehat{q}_u(c^*)}{\widehat{q}_p(c^*)} + \left(\frac{\gamma}{\widehat{q}_p(c)} - \frac{1}{\widehat{q}_p(c^*)}\right)\left(\sqrt{\frac{\log(4/\delta)}{2n_u}} + \sqrt{\frac{\log(4/\delta)}{2n_p}}\right) \tag{12}$$

Using (12) at $c = c'$, we have the following lower bound on ucb at $c'$:

$$\frac{\widehat{q}_u(c')}{\widehat{q}_p(c')} + \frac{1+\gamma}{\widehat{q}_p(c')}\left(\sqrt{\frac{\log(4/\delta)}{2n_u}} + \sqrt{\frac{\log(4/\delta)}{2n_p}}\right) \tag{13}$$

$$\geq \frac{\widehat{q}_u(c^*)}{\widehat{q}_p(c^*)} + \frac{1+\gamma}{\widehat{q}_p(c^*)}\left(\sqrt{\frac{\log(4/\delta)}{2n_u}} + \sqrt{\frac{\log(4/\delta)}{2n_p}}\right), \tag{14}$$

Moreover from (12), we also have that the lower bound on ucb at $c \geq c'$ is strictly greater than the lower bound on ucb at $c'$. Using definition of $\widehat{c}$, we have

$$\frac{\widehat{q}_u(c^*)}{\widehat{q}_p(c^*)} + \frac{1+\gamma}{\widehat{q}_p(c^*)}\left(\sqrt{\frac{\log(4/\delta)}{2n_u}} + \sqrt{\frac{\log(4/\delta)}{2n_p}}\right) \tag{15}$$

$$\geq \frac{\widehat{q}_u(\widehat{c})}{\widehat{q}_p(\widehat{c})} + \frac{1+\gamma}{\widehat{q}_p(\widehat{c})}\left(\sqrt{\frac{\log(4/\delta)}{2n_u}} + \sqrt{\frac{\log(4/\delta)}{2n_p}}\right), \tag{16}$$

and hence

$$\widehat{c} \leq c'. \tag{17}$$

**Part 2:** We now establish an upper and lower bound on $\widehat{\alpha}$. We start with upper confidence bound on $\widehat{\alpha}$. By definition of $\widehat{c}$, we have

$$\frac{\widehat{q}_u(\widehat{c})}{\widehat{q}_p(\widehat{c})} + \frac{1+\gamma}{\widehat{q}_p(\widehat{c})}\left(\sqrt{\frac{\log(4/\delta)}{2n_u}} + \sqrt{\frac{\log(4/\delta)}{2n_p}}\right) \tag{18}$$

$$\leq \min_{c \in [0,1]}\left[\frac{\widehat{q}_u(c)}{\widehat{q}_p(c)} + \frac{1+\gamma}{\widehat{q}_p(c)}\left(\sqrt{\frac{\log(4/\delta)}{2n_u}} + \sqrt{\frac{\log(4/\delta)}{2n_p}}\right)\right] \tag{19}$$

$$\leq \frac{\widehat{q}_u(c^*)}{\widehat{q}_p(c^*)} + \frac{1+\gamma}{\widehat{q}_p(c^*)}\left(\sqrt{\frac{\log(4/\delta)}{2n_u}} + \sqrt{\frac{\log(4/\delta)}{2n_p}}\right). \tag{20}$$

Using Lemma 1 at $c^*$, we get

$$\frac{\widehat{q}_u(c^*)}{\widehat{q}_p(c^*)} \leq \frac{q_u(c^*)}{q_p(c^*)} + \frac{1}{\widehat{q}_p(c^*)}\left(\sqrt{\frac{\log(4/\delta)}{2n_u}} + \frac{q_u(c^*)}{q_p(c^*)}\sqrt{\frac{\log(4/\delta)}{2n_p}}\right)$$

$$= \alpha^* + \frac{1}{\widehat{q}_p(c^*)}\left(\sqrt{\frac{\log(4/\delta)}{2n_u}} + \alpha^*\sqrt{\frac{\log(4/\delta)}{2n_p}}\right). \tag{21}$$

Combining (20) and (21), we get

$$\widehat{\alpha} = \frac{\widehat{q}_u(\widehat{c})}{\widehat{q}_p(\widehat{c})} \leqslant \alpha^* + \frac{2+\gamma}{\widehat{q}_p(c^*)} \left( \sqrt{\frac{\log(4/\delta)}{2n_u}} + \sqrt{\frac{\log(4/\delta)}{2n_p}} \right) . \tag{22}$$

Using DKW inequality on $\widehat{q}_p(c^*)$, we have $\widehat{q}_p(c^*) \geqslant q_p(c^*) - \sqrt{\frac{\log(4/\delta)}{2n_p}}$. Assuming $n_p \geqslant \frac{2\log(4/\delta)}{q_p^2(c^*)}$, we get $\widehat{q}_p(c^*) \leqslant q_p(c^*)/2$ and hence,

$$\widehat{\alpha} \leqslant \alpha^* + \frac{4+2\gamma}{q_p(c^*)} \left( \sqrt{\frac{\log(4/\delta)}{2n_u}} + \sqrt{\frac{\log(4/\delta)}{2n_p}} \right) . \tag{23}$$

Finally, we now derive a lower bound on $\widehat{\alpha}$. From Lemma 1, we have the following inequality at $\widehat{c}$

$$\frac{q_u(\widehat{c})}{q_p(\widehat{c})} \leqslant \frac{\widehat{q}_u(\widehat{c})}{\widehat{q}_p(\widehat{c})} + \frac{1}{\widehat{q}_p(\widehat{c})} \left( \sqrt{\frac{\log(4/\delta)}{2n_u}} + \frac{q_u(\widehat{c})}{q_p(\widehat{c})}\sqrt{\frac{\log(4/\delta)}{2n_p}} \right) . \tag{24}$$

Since $\alpha^* \leqslant \frac{q_u(\widehat{c})}{q_p(\widehat{c})}$, we have

$$\alpha^* \leqslant \frac{q_u(\widehat{c})}{q_p(\widehat{c})} \leqslant \frac{\widehat{q}_u(\widehat{c})}{\widehat{q}_p(\widehat{c})} + \frac{1}{\widehat{q}_p(\widehat{c})} \left( \sqrt{\frac{\log(4/\delta)}{2n_u}} + \frac{q_u(\widehat{c})}{q_p(\widehat{c})}\sqrt{\frac{\log(4/\delta)}{2n_p}} \right) . \tag{25}$$

Using (23), we obtain a very loose upper bound on $\frac{\widehat{q}_u(\widehat{c})}{\widehat{q}_p(\widehat{c})}$. Assuming $\min(n_p, n_u) \geqslant \frac{2\log(4/\delta)}{q_p^2(c^*)}$, we have $\frac{\widehat{q}_u(\widehat{c})}{\widehat{q}_p(\widehat{c})} \leqslant \alpha^* + 4 + 2\gamma \leqslant 5 + 2\gamma$. Using this in (25), we have

$$\alpha^* \leqslant \frac{\widehat{q}_u(\widehat{c})}{\widehat{q}_p(\widehat{c})} + \frac{1}{\widehat{q}_p(\widehat{c})} \left( \sqrt{\frac{\log(4/\delta)}{2n_u}} + (5+2\gamma)\sqrt{\frac{\log(4/\delta)}{2n_p}} \right) . \tag{26}$$

Moreover, as $\widehat{c} \geqslant c'$, we have $\widehat{q}_p(\widehat{c}) \geqslant \frac{\gamma}{2+\gamma}\widehat{q}_p(c^*)$ and hence,

$$\alpha^* - \frac{\gamma+2}{\gamma\widehat{q}_p(c^*)} \left( \sqrt{\frac{\log(4/\delta)}{2n_u}} + (5+2\gamma)\sqrt{\frac{\log(4/\delta)}{2n_p}} \right) \leqslant \frac{\widehat{q}_u(\widehat{c})}{\widehat{q}_p(\widehat{c})} = \widehat{\alpha} . \tag{27}$$

As we assume $n_p \geqslant \frac{2\log(4/\delta)}{q_p^2(c^*)}$, we have $\widehat{q}_p(c^*) \leqslant q_p(c^*)/2$, which implies the following lower bound on $\alpha$:

$$\alpha^* - \frac{2\gamma+4}{\gamma q_p(c^*)} \left( \sqrt{\frac{\log(4/\delta)}{2n_u}} + (5+2\gamma)\sqrt{\frac{\log(4/\delta)}{2n_p}} \right) \leqslant \widehat{\alpha} . \tag{28}$$

$\square$

*Proof of Corollary 1.* Note that since $\alpha \leqslant \alpha^*$, the lower bound remains the same as in Theorem 1. For upper bound, plugging in $q_u(c) = \alpha q_p(c) + (1-\alpha)q_n(c)$, we have $\alpha^* = \alpha + (1-\alpha)q_n(c^*)/q_p(c^*)$ and hence, the required upper bound. $\square$

## B.1 Note on $\gamma$ in Algorithm 1

We multiply the upper bound in Lemma 1 to establish lower bound on $\widehat{q}_p(\widehat{c})$. Otherwise, in an extreme case, with $\gamma = 0$, Algorithm 1 can select $\widehat{c}$ with arbitrarily low $\widehat{q}_p(\widehat{c})$ ($\ll q_p(c^*)$) and hence poor concentration guarantee to the true mixture proportion. However, with a small positive $\gamma$, we can obtain lower bound on $\widehat{q}_p(\widehat{c})$ and hence tight guarantees on the ratio estimate ($\widehat{q}_u(\widehat{c})/\widehat{q}_p(\widehat{c})$) in Theorem 1.

In our experiments, we choose $\gamma = 0.01$. However, we didn't observe any (significant) differences in mixture proportion estimation even with $\gamma = 0$. implying that we never observe $\widehat{q}_p(\widehat{c})$ taking arbitrarily small values in our experiments.

| Dataset | Model | $(\text{TED})^n$ | BBE* | DEDPUL* | Scott* |
|---|---|---|---|---|---|
| Binarized CIFAR | ResNet | **0.018** | 0.072 | 0.075 | 0.091 |
| CIFAR Dog vs Cat | ResNet | **0.074** | 0.120 | 0.113 | 0.158 |
| Binarized MNIST | MLP | **0.021** | 0.028 | 0.027 | 0.063 |
| MNIST17 | MLP | **0.003** | 0.008 | 0.006 | 0.037 |

Table 3: Absolute estimation error when $\alpha$ is 0.5. "*" denote oracle early stopping as defined in Sec. 6. As mentioned in Scott [39] implementation in https://web.eecs.umich.edu/~cscott/code/mpe_v2.zip, we use the binomial inversion at $\delta$ instead of $\delta/n$ (rescaling using the union bound). Since we are using Binomial inversion at n discrete points simultaneously, we should use the union-bound penalty. However, using union bound penalty substantially increases the bias in their estimator.

## C   Comparison of BBE with Scott [39]

Heuristic estimator due to Scott [39] is motivated by the estimator in Blanchard et al. [6]. The estimator in Blanchard et al. [6] relies on VC bounds, which are known to be loose in typical deep learning situations. Therefore, Scott [39] proposed an heuristic implementation based on the minimum slope of any point in the ROC space to the point $(1, 1)$. To obtain ROC estimates, authors use direct binomial tail inversion (instead of VC bounds as in Blanchard et al. [6]) to obtain tight upper bounds for true positives and lower bounds for true negatives. Finally, using these conservatives estimates the estimator in Scott [39] is obtained as the minimum slope of any of the operating points to the point $(1, 1)$.

While the estimate of one minus true positives at a threshold $t$ is similar in spirit to our number of unlabeled examples in the top bin and the estimate of one minus true negatives at a threshold $t$ is similar in spirit to our number of positive examples in the unlabeled data, the functional form of these estimates are very different. Scott [39] estimator is the ratio of quantities obtained by binomial tail inversion (i.e. upper bound in the numerator and lower bound in the denominator). By contrast, the final BBE estimate is simply the ratio of empirical CDFs at the optimal threshold. Mathematically, we have

$$\widehat{\alpha}_{\text{Scott}} = \frac{\widehat{q}_u(c_{\text{Scott}}) + \text{binv}(n_u, \widehat{q}_u(c_{\text{Scott}}), \delta/n_u)}{\widehat{q}_p(c_{\text{Scott}}) - \text{binv}(n_p, \widehat{q}_p(c_{\text{Scott}}), \delta/n_p)} \quad \text{and} \quad (29)$$

$$\widehat{\alpha}_{\text{BBE}} = \frac{\widehat{q}_u(c_{\text{BBE}})}{\widehat{q}_p(c_{\text{BBE}})}, \quad (30)$$

where $c_{\text{Scott}} = \arg\min_{c \in [0,1]} \frac{\widehat{q}_u(c) + \text{binv}(n_u, \widehat{q}_u(c), \delta/n_u)}{\widehat{q}_p(c) - \text{binv}(n_p, \widehat{q}_p(c), \delta/n_p)}$ and $\text{binv}(n_p, q_p(c), \delta/n_p)$ is the tightest possible deviation bound for a binomial random variable [39] and and $c_{\text{BBE}}$ is given by Algorithm 1. Moreover, Scott [39] provide no theoretical guarantees for their heuristic estimator $\widehat{\alpha}_{\text{Scott}}$. On the hand, we provide guarantees that our estimator $\widehat{\alpha}_{\text{BBE}}$ will converge to the best estimate achievable over all choices of the bin size and provide consistent estimates whenever a pure top bin exists. Supporting theoretical results of BBE, we observe that these choices in BBE create substantial differences in the empirical performance as observed in Table 3. We repeat experiment for MPE from Sec. 6 where we compare other methods with the Scott [39] estimator as defined in (29).

As a side note, a naive implementation of $\widehat{\alpha}_{\text{Scott}}$ instead of (29) where we directly minimize the empirical ratio yields poor estimates due to noise introduced with finite samples. In our experiments, we observed that $\widehat{\alpha}_{\text{Scott}}$ improves a lot over this naive estimator.

## D   Toy setup

Jain et al. [21] and Ivanov [20] discuss Bayes optimality of the PvU classifier (or its one-to-one mapping) as a sufficient condition to preserve $\alpha$ in transformed space. However, in a simple toy setup (in App. D), we show that even when the hypothesis class is well specified for PvN learning, it will not in general contain the Bayes optimal scoring function for PvU data and thus PvU training will not recover the Bayes-optimal scoring function, even in population.

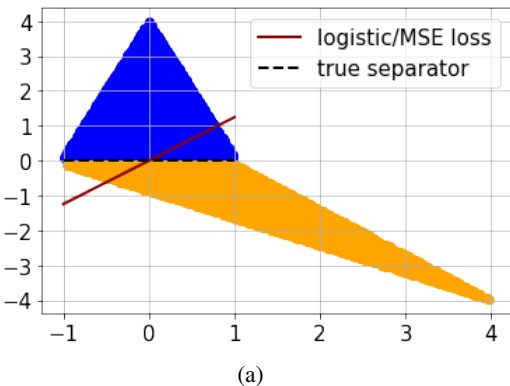

(a)

Figure 4: Blue points show samples from the positive distribution and orange points show samples from the negative distribution. Unalabeled data is obtained by mixing positive and negative distribution with equal proportion. BCE (or Brier) loss minimization on P vs U data leads to a classifiers that is not consistent with the ranking of the Bayes optimal score function.

Consider a scenario with $\mathcal{X} = \mathbb{R}^2$. Assume points from the positive class are sampled uniformly from the interior of the triangle defined by coordinates $\{(-1, 0.1), (0, 4), (1, 0.1)\}$ and negative points are sampled uniformly from the interior of triangle defined by coordinates $\{(-1, -0.1), (4, -4), (1, -0.1)\}$. Ref. to Fig. 4 for a pictorial representation. Let mixture proportion be $0.5$ for the unlabeled data. Given access to distribution of positive data and unlabeled data, we seek to train a linear classifier to minimize logistic or Brier loss for PvU training.

Since we need a monotonic transformation of the Bayes optimal scoring function, we want to recover a predictor parallel to x-axis, the Bayes optimal classifier for PvN training. However, minimizing the logistic loss (or Brier loss) using numerical methods, we obtain a predictor that is inclined at a non-zero acute angle to the x-axis. Thus, the PvU classifier obtained fails to satisfy the sufficient condition from Jain et al. [21] and Ivanov [20]. On the other hand, note that the linear classifier obtained by PvU training satisfies the pure positive bin property.

Now we show that under the subdomain assumption [39, 35], any monotonic transformation of Bayes optimal scoring function induces positive pure bin property. First, we define the subdomain assumption.

**Assumption 1** (Subdomain assumption). *A family of subsets $\mathcal{S} \subseteq 2^{\mathcal{X}}$, and distributions $p_p$, $p_n$ are said to satisfy the anchor set condition with margin $\gamma > 0$, if there exists a compact set $A \in \mathcal{S}$ such that $A \subseteq supp(p_p)/supp(p_n)$ and $p_p(A) \geqslant \gamma$.*

Note that any monotonic mapping of the Bayes optimal scoring function can be represented by $\tau' = g \circ \tau$, where g is a monotonic function and

$$\tau(x) = \begin{cases} p_p(x)/p_u(x) & \text{if } p_p(x) > 0 \\ 0 & \text{o.w} . \end{cases} \tag{31}$$

For any point $x \in A$ and $x' \in \mathcal{X}/A$, we have $\tau(x) > \tau(x')$ which implies $\tau'(x) > \tau'(x')$. Thus, any monotonic mapping of Bayes optimal scoring function yields the positive pure bin property with $\epsilon_p \geqslant \gamma$.

## E    Analysis of CVIR

First we analyse our loss function in the scenario when the support of positives and negatives is separable. We assume that the true alpha $\alpha$ is known and we have access to populations of positive and unlabeled data. We also assume that their exists a separator $f^* : \mathcal{X} \mapsto \{0, 1\}$ that can perfectly separate the positive and negative distribution, i.e., $\int dx p_p(x) \mathbb{I}\left[f^*(x) \neq 1\right] + \int dx p_n(x) \mathbb{I}\left[f^*(x) \neq 0\right] = 0$. Our learning objective can be written as jointly optimizing a classifier $f$ and a weighting function $w$

on the unlabeled distribution:

$$\min_{f \in \mathcal{F}, w} \int dx p_p(x) l(f(x), 1) + \frac{1}{1 - \alpha} \int dx p_u(x) w(x) l(f(x), 0),$$

$$\text{s.t. } w : \mathcal{X} \mapsto [0, 1], \int dx p_u(x) w(x) = 1 - \alpha. \tag{32}$$

The following proposition shows that minimizing the objective (32) on separable positive and negative distributions gives a perfect classifier.

**Proposition 1.** *For $\alpha \in (0, 1)$, if there exists a classifier $f^* \in \mathcal{F}$ that can perfectly separate the positive and negative distributions, optimizing objective (32) with 0-1 loss leads to a classifier $f$ that achieves $0$ classification error on the unlabeled distribution.*

*Proof.* First we observe that having $w(x) = 1 - f^*(x)$ leads to the objective value being minimized to 0 as well as a perfect classifier $f$. This is because

$$\frac{1}{1 - \alpha} \int dx p_u(x)(1 - f^*(x)) l(f(x), 0) = \int dx p_n(x) l(f(x), 0)$$

thus the objective becomes classifying positive v.s. negative, which leads to a perfect classifier if $\mathcal{F}$ contains one. Now we show that for any $f$ such that the classification error is non-zero then the objective (32) must be greater than zero no matter what $w$ is. Suppose $f$ satisfies

$$\int dx p_p(x) l(f(x), 1) + \int dx p_n(x) l(f(x), 0) > 0.$$

We know that either $\int dx p_p(x) l(f(x), 1) > 0$ or $\int dx p_n(x) l(f(x), 0) > 0$ will hold. If $\int dx p_p(x) l(f(x), 1) > 0$ we know that (32) must be positive. If $\int dx p_p(x) l(f(x), 1) = 0$ and $\int dx p_n(x) l(f(x), 0) > 0$ we have $l(f(x), 0) = 1$ almost everywhere in $p_p(x)$ thus

$$\frac{1}{1 - \alpha} \int dx p_u(x) w(x) l(f(x), 0)$$

$$= \frac{\alpha}{1 - \alpha} \int dx p_p(x) w(x) l(f(x), 0) + \int dx p_n(x) w(x) l(f(x), 0)$$

$$= \frac{\alpha}{1 - \alpha} \int dx p_p(x) w(x) + \int dx p_n(x) w(x) l(f(x), 0).$$

If $\int dx p_p(x) w(x) > 0$ we know that (32) must be positive. If $\int dx p_p(x) w(x) = 0$, since we know that

$$\int dx p_u(x) w(x) = \alpha \int dx p_p(x) w(x) + (1 - \alpha) \int dx p_n(x) w(x) = 1 - \alpha$$

we have $\int dx p_n(x) w(x) = 1$ which means $w(x) = 1$ almost everywhere in $p_n(x)$. This leads to the fact that $\int dx p_n(x) l(f(x), 0) > 0$ indicates $\int dx p_n(x) w(x) l(f(x), 0) > 0$, which concludes the proof.

$\square$

The intuition is that, any classifier that discards an $\tilde{\alpha} > 0$ proportion of negative distribution from unlabeled will have loss strictly greater than zero with our CVIR objective. Since only a perfect linear separator (with weights $\to \infty$) can achieves loss $\to 0$, CVIR objective will (correctly) discard the $\alpha$ proportion of positive from unlabeled data achieving a classifier that perfectly separates the data.

We leave theoretic investigation on non-separable distributions for future work. However, as an initial step towards a general theory, we show that in the population case one step of our alternating procedure cannot increase the loss.

Consider the following objective function

$$L(f_t, w_t) = E_{x \sim P_p}[l(f_t(x), 0)] + E_{x \sim P_u}[w_t(x) l(f_t(x), 1)] \tag{33}$$

$$\text{such that} \quad E_{x \sim P_u}[w(x)] = 1 - \alpha \text{ and } w(x) \in \{0, 1\}$$

Given $f_t$ and $w_t$, CVIR can be summarized as the following two step iterative procedure: (i) Fix $f_t$, optimize the loss to obtain $w_{t+1}$; and (ii) Fix $w_{t+1}$ and optimize the loss to obtain $f_{t+1}$. By construction of CVIR, we select $w_{t+1}$ such that we discard points with highest loss, and hence $L(f_t, w_{t+1}) \leqslant L(f_t, w_t)$. Fixing $w_{t+1}$, we minimize the $L(f_t, w_{t+1})$ to obtain $f_{t+1}$ and hence $L(f_{t+1}, w_{t+1}) \leqslant L(f_t, w_{t+1})$. Combining these two steps, we get $L(f_{t+1}, w_{t+1}) \leqslant L(f_t, w_t)$.

## F  Experimental Details

Below we present dataset details. We present experiments with MNIST Overlap in App. G.8.

| Dataset | Simulated PU Dataset | P vs N | #Positives | | #Unlabeled | |
|---|---|---|---|---|---|---|
| | | | Train | Val | Train | Val |
| CIFAR10 | Binarized CIFAR | [0-4] vs [5-9] | 12500 | 12500 | 2500 | 2500 |
| | CIFAR Dog vs Cat | 3 vs 5 | 2500 | 2500 | 500 | 500 |
| MNIST | Binarized MNIST | [0-4] vs [5-9] | 15000 | 15000 | 2500 | 2500 |
| | MNIST 17 | 1 vs 7 | 3000 | 3000 | 500 | 500 |
| | MNIST Overlap | [0-7] vs [3-9] | 150000 | 15000 | 2500 | 2500 |
| IMDb | IMDb | pos vs neg | 6250 | 6250 | 5000 | 5000 |

For CIFAR dataset, we also use the standard data augemention of random crop and horizontal flip. PyTorch code is as follows:

```
(transforms.RandomCrop(32, padding=4),
transforms.RandomHorizontalFlip())
```

### F.1  Architecture and Implementation Details

All experiments were run on NVIDIA GeForce RTX 2080 Ti GPUs. We used PyTorch [33] and Keras with Tensorflow [1] backend for experiments.

For CIFAR10, we experiment with convolutional nets and MLP. For MNIST, we train MLP. In particular, we use ResNet18 [19] and all convolution net [40] . Implementation adapted from: `https://github.com/kuangliu/pytorch-cifar.git`. We consider a 4-layered MLP. The PyTorch code for 4-layer MLP is as follows:

```
 nn.Sequential(nn.Flatten(),
nn.Linear(input_dim, 5000, bias=True),
nn.ReLU(),
nn.Linear(5000, 5000, bias=True),
nn.ReLU(),
nn.Linear(5000, 50, bias=True),
nn.ReLU(),
nn.Linear(50, 2, bias=True)
)
```

For all architectures above, we use Xaviers initialization [18]. For all methods except nnPU and uPU, we do cross entropy loss minimization with SGD optimizer with momentum $0.9$. For convolution architectures we use a learning rate of $0.1$ and MLP architectures we use a learning rate of $0.05$. For nnPU and uPU, we minimize sigmoid loss with ADAM optimizer with learning rate $0.0001$ as advised in its original paper. For all methods, we fix the weight decay param at $0.0005$.

For IMDb dataset, we fine-tune an off-the-shelf uncased BERT model [10]. Code adapted from Hugging Face Transformers [42]: `https://huggingface.co/transformers/v3.1.0/custom_datasets.html`. For all methods except nnPU and uPU, we do cross entropy loss minimization

with Adam optimizer with learning rate 0.00005 (default params). With the same hyperparameters and Sigmoid loss, we could not train BERT with nnPU and uPU due to vanishing gradients. Instead we use learning rate 0.00001.

## F.2 Division between training set and hold-out set

Since the training set is used to learn the classifier (parameters of a deep neural network) and the hold-out set is just used to learn the mixture proportion estimate (scalar), we use a larger dataset for training. Throughout the experiments, we use an 80-20 split of the original set.

At a high level, we have an error bound on the mixture proportion estimate and we can use that to decide the split in general. As long as we use enough samples to make the $\mathcal{O}(1/\sqrt{n})$ small in our bound in Theorem 1, we can use the rest of the samples to learn the classifier.

# G  Additional Experiments

## G.1  nnPU vs PN classification

In this section, we compare the performance of nnPU and PvN training on the same positive and negative (from the unlabeled) data at $\alpha = 0.5$. We highlight the huge classification performance gap between nnPU and PvN training and show that training with CVuO objective partially recovers the performance gap. Note, to train PvN classifier, we use the same hyperparameters as that with PvU training.

| Dataset | Model | nnPU (known $\alpha$) | PvN | CVuO (known $\alpha$) | (TED)$^n$ (unknown $\alpha$) |
|---------|-------|------|-----|------|------|
| Binarized CIFAR | ResNet | 76.8 | 86.9 | 82.6 | 82.7 |
| | All Conv | 72.1 | 76.7 | 77.1 | 76.8 |
| | MLP | 63.9 | 65.1 | 65.9 | 63.2 |
| CIFAR Dog vs Cat | ResNet | 72.6 | 80.4 | 74.0 | 76.1 |
| | All Conv | 68.4 | 77.9 | 71.0 | 72.2 |
| Binarized MNIST | MLP | 95.9 | 96.7 | 96.4 | 95.9 |
| MNIST17 | MLP | 98.2 | 99.0 | 98.6 | 98.6 |
| IMDb | BERT | 86.2 | 89.1 | 87.4 | 88.1 |

Table 4: Accuracy for PvN classification with nnPU, PvN, CVuO objective and (TED)$^n$ training. Results reported by aggregating aggregating over 10 epochs.

## G.2  Under-Fitting due to pessimistic early stopping

Ivanov [20] explored the following heuristics for ad-hoc early stopping criteria: training proceeds until the loss on unseen PU data ceases to decrease. In particular, the authors suggested early stopping criterion based on the loss on unseen PU data doesn't decrease in epochs separated by a pre-defined window of length $l$. The early stopping is done when this happens consecutively for $l$ epochs. However, this approach leads to severe under-fitting. When we fix $l = 5$, we observe a significant performance drop in CIFAR classification and MPE.

With PvU training, the performance of ResNet model on Binarized CIFAR (in Table 2) drops from 78.3 (orcale stopping) to 60.4 (with early stopping). Similar on CIFAR CAT vs Dog, the performance of the same architecture drops from 71.6 (orcale stopping) to 58.4 (with early stopping). Note that the decrease in accuracy is less or not significant for MNIST. With PvU training, the performance of MLP model on Binarized MNIST (in Table 2) drops from 94.5 (orcale stopping) to 94.1 (with early stopping). This is because we obtain good performance on MNIST early in training.

### G.3 Results parallel to Fig. 3

Epoch wise results for all models for Binarized CIFAR, CIFAR Dog vs Cat, Binarized MNIST, MNIST 17 and IMDb.

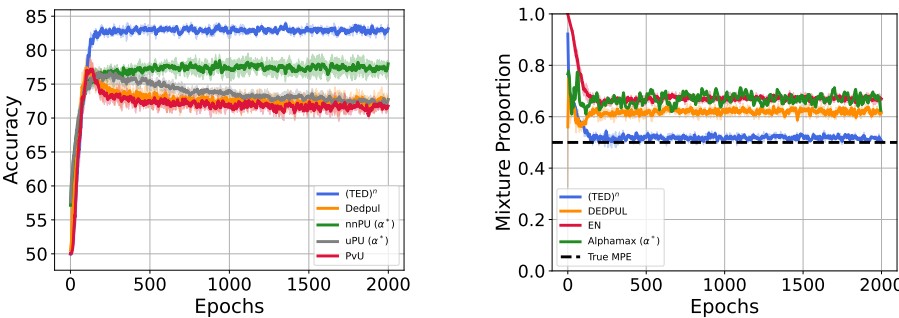

Figure 5: Epoch wise results with ResNet-18 network trained on CIFAR-binarized.

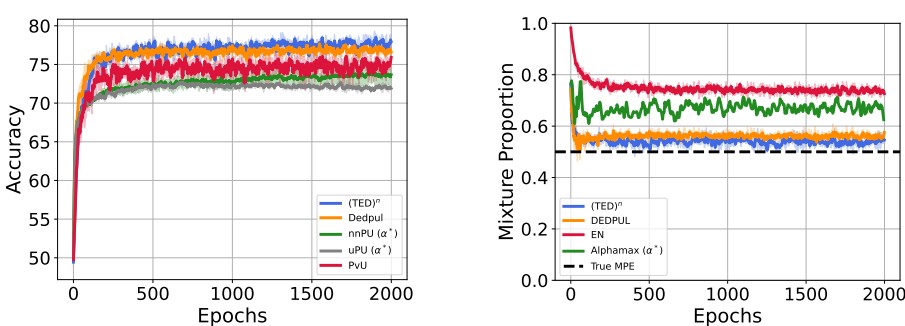

Figure 6: Epoch wise results with All convolutional network trained on CIFAR-binarized.

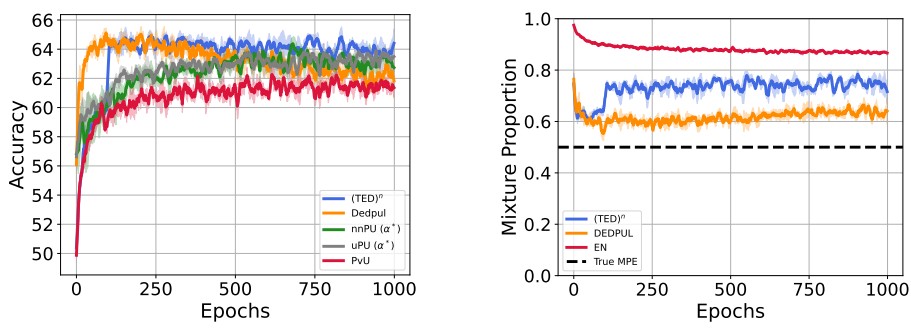

Figure 7: Epoch wise results with FCN trained on CIFAR-binarized.

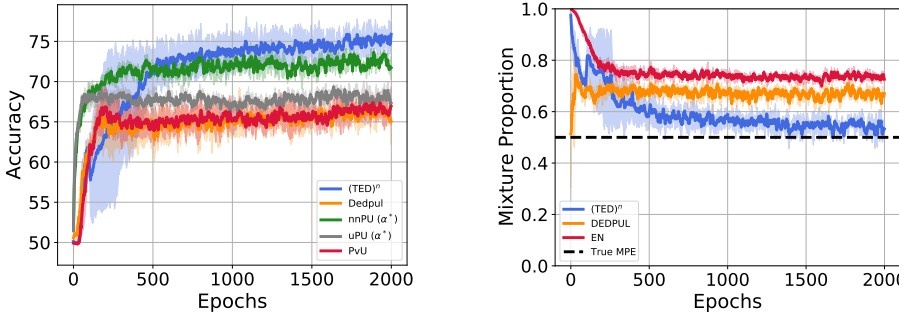

Figure 8: Epoch wise results with ResNet-18 trained on CIFAR Dog vs Cat.

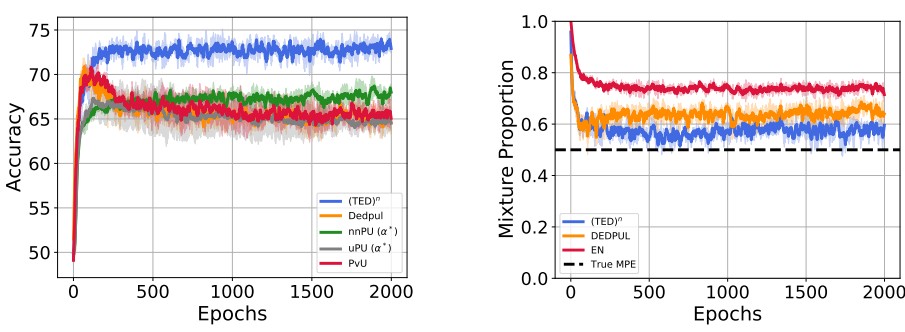

Figure 9: Epoch wise results with All convolutional network trained on CIFAR Dog vs Cat.

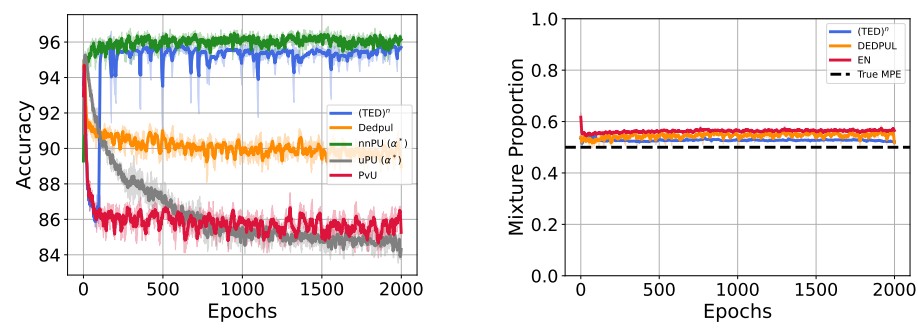

Figure 10: Epoch wise results with MLP trained on Binarized MNIST.

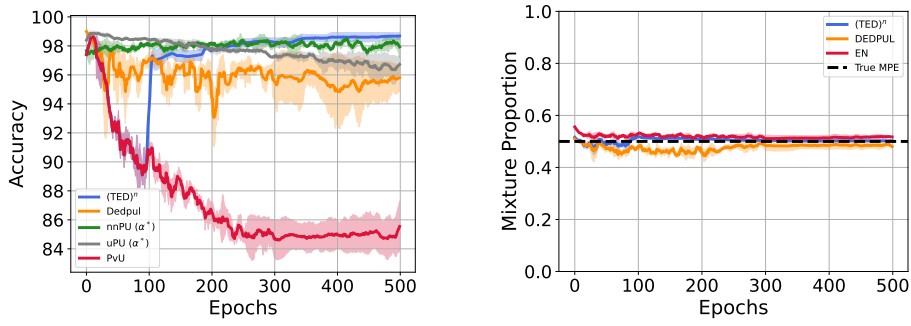

Figure 11: Epoch wise results with MLP trained on MNIST 17.

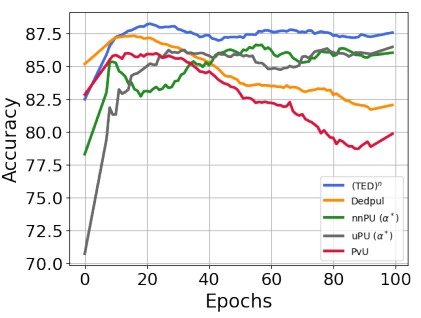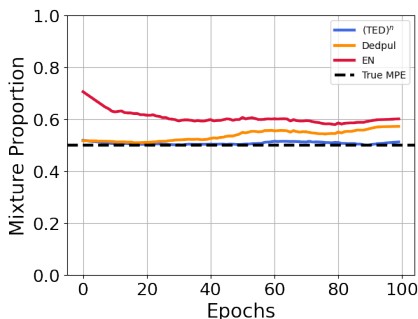

Figure 12: Epoch wise results with BERT trained on IMDb.

## G.4 Overfitting on unlabeled data as PvU training proceeds

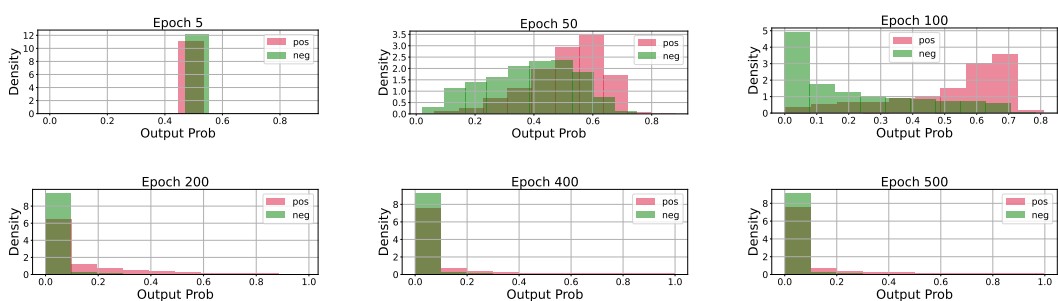

Figure 13: Score assigned by the classifier to positive and negative points in the unlabeled training set as PvU training proceeds. As training proceeds, classifier memorizes both positive and negative in unlabeled as negatives.

In Fig. 13, we show the distribution of unlabeled training points. We show that as positive versus unlabeled training proceeds with a ResNet-18 model on binarized CIFAR dataset, classifier memorizes all the unlabeled data as negative assigning them very small scores (i.e., the probability of them being negative).

## G.5 Ablations to $(TED)^n$

**Varying the number of warm start epochs** We now vary the number of warm start epochs with $(TED)^n$. We observe that increasing the number of warm start epochs doesn't hurt $(TED)^n$ even when the classifier at the end of the warm start training memorized PU training data due PvU training. While in many cases $(TED)^n$ training without warm start is able to recover the same performance, it fails to learn anything for CIFAR Dog vs Cat with all convolutional neural network. This highlights the need for warm start training with $(TED)^n$.

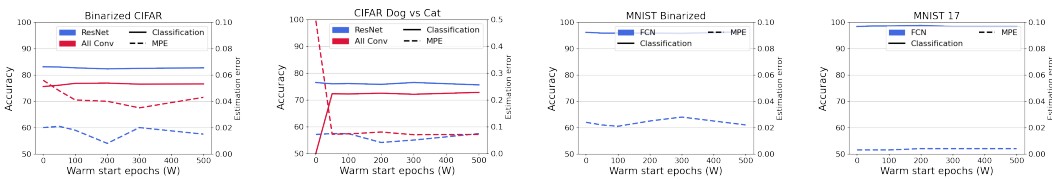

Figure 14: Classification and MPE results with varying warm start epochs $W$ with $(TED)^n$

**Varying the true mixture proportion $\alpha$** Next, we vary $\alpha$, the true mixture proportion and present results for MPE and classification in Fig. 15. Overall, across all $\alpha$, our method $(TED)^n$ is able to

achieve superior performance as compared to alternate algorithms. We omit high $\alpha$ for CIFAR and IMDb datasets as all the methods result in trivial accuracy and mixture proportion estimate.

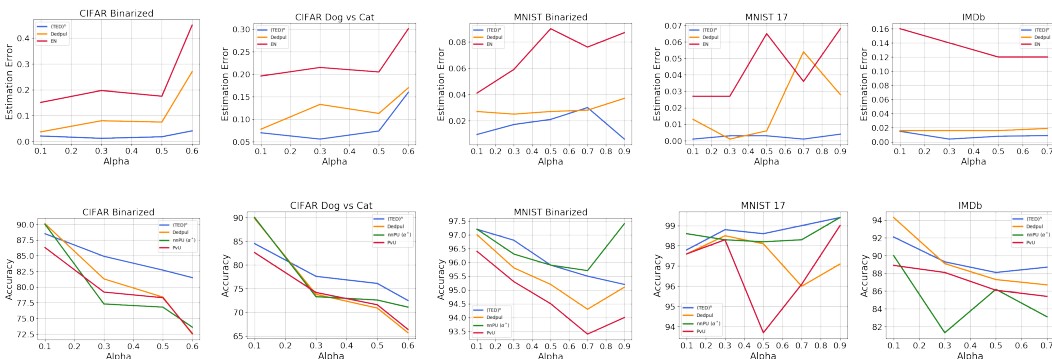

Figure 15: MPE and Classification results with varying mixture proportion. For each method we show results with the best performing architecture.

## G.6  Classification and MPE results with error bars

| Dataset | Model | $(TED)^n$ | BBE* | DEDPUL* | EN | KM2 | TiCE |
|---|---|---|---|---|---|---|---|
| Binarized CIFAR | ResNet | **0.026 ± 0.005** | 0.091 ± 0.027 | 0.091 ± 0.023 | 0.192 ± 0.007 | | |
| | All Conv | 0.042 ± 0.003 | **0.037 ± 0.018** | 0.052 ± 0.017 | 0.221 ± 0.017 | 0.168 ± 0.207 | 0.194 ± 0.039 |
| | MLP | 0.225 ± 0.013 | 0.177 ± 0.011 | **0.138 ± 0.009** | 0.372 ± 0.002 | | |
| CIFAR Dog vs Cat | ResNet | **0.078 ± 0.010** | 0.176 ± 0.015 | 0.170 ± 0.010 | 0.226 ± 0.003 | 0.331 ± 0.238 | 0.286 ± 0.013 |
| | All Conv | **0.066 ± 0.015** | 0.128 ± 0.020 | 0.115 ± 0.014 | 0.250 ± 0.019 | | |
| Binarized MNIST | MLP | **0.024 ± 0.001** | 0.032 ± 0.001 | 0.031 ± 0.003 | 0.080 ± 0.009 | 0.029 ± 0.008 | 0.056 ± 0.05 |
| MNIST17 | MLP | **0.003 ± 0.000** | 0.023 ± 0.017 | 0.021 ± 0.011 | 0.028 ± 0.017 | 0.022 ± 0.003 | 0.043 ± 0.023 |
| IMDb | BERT | **0.008 ± 0.001** | 0.011 ± 0.002 | 0.016 ± 0.005 | 0.07 ± 0.01 | - | - |

Table 5: Absolute estimation error when $\alpha$ is 0.5. "*" denote oracle early stopping as defined in Sec. 6. Results reported by aggregating absolute error over 10 epochs and 3 seeds.

| Dataset | Model | $(TED)^n$ (unknown $\alpha$) | CVIR (known $\alpha$) | PvU* (known $\alpha$) | DEDPUL* (unknown $\alpha$) | nnPU (known $\alpha$) | uPU* (known $\alpha$) |
|---|---|---|---|---|---|---|---|
| Binarized CIFAR | ResNet | **82.7 ± 0.13** | 82.3 ± 0.18 | 76.9 ± 1.12 | 77.1 ± 1.52 | 77.2 ± 1.03 | 76.7 ± 0.74 |
| | All Conv | 77.9 ± 0.29 | **78.1 ± 0.47** | 75.8 ± 0.75 | 77.1 ± 0.64 | 73.4 ± 1.31 | 72.5 ± 0.21 |
| | MLP | 64.2 ± 0.37 | **66.9 ± 0.28** | 61.6 ± 0.38 | 62.6 ± 0.30 | 63.1 ± 0.79 | 64.0 ± 0.24 |
| CIFAR Dog vs Cat | ResNet | **75.2 ± 1.74** | 73.3 ± 0.94 | 67.3 ± 1.52 | 67.0 ± 1.46 | 71.8 ± 0.33 | 68.8 ± 0.53 |
| | All Conv | **73.0 ± 0.81** | 71.7 ± 0.47 | 70.5 ± 0.60 | 69.2 ± 0.86 | 67.9 ± 0.52 | 67.5 ± 2.28 |
| Binarized MNIST | MLP | 95.6 ± 0.42 | **96.3 ± 0.07** | 94.2 ± 0.58 | 94.8 ± 0.10 | 96.1 ± 0.14 | 95.2 ± 0.19 |
| MNIST17 | MLP | **98.7 ± 0.25** | **98.7 ± 0.09** | 96.9 ± 1.51 | 97.7 ± 0.62 | 98.4 ± 0.20 | 98.4 ± 0.09 |
| IMDb | BERT | **87.6 ± 0.20** | 87.4 ± 0.25 | 86.1 ± 0.53 | 87.3 ± 0.18 | 86.2 ± 0.25 | 85.9 ± 0.12 |

Table 6: Accuracy for PvN classification with PU learning. "*" denote oracle early stopping as defined in Sec. 6. Results reported by aggregating over 10 epochs and 3 seeds.

## G.7 Experiments on UCI dataset

In this section, we will present results on 5 UCI datasets.

| Dataset | #Positives | | #Unlabeled | |
|---|---|---|---|---|
| | Train | Val | Train | Val |
| concrete | 162 | 162 | 81 | 81 |
| mushroom | 1304 | 1304 | 652 | 652 |
| landsat | 946 | 946 | 472 | 472 |
| pageblock | 185 | 185 | 92 | 92 |
| spambase | 604 | 604 | 302 | 302 |

We train a MLP with 2 hidden layers each with $512$ units. The PyTorch code for 4-layer MLP is as follows:

```
 nn.Sequential(nn.Flatten(),
nn.Linear(input_dim, 512, bias=True),
nn.ReLU(),
nn.Linear(512, 512, bias=True),
nn.ReLU(),
nn.Linear(512, 2, bias=True),
)
```

Similar to vision datasets and architectures, we do cross entropy loss minimization with SGD optimizer with momentum 0.9 and learning rate 0.1. For nnPU and uPU, we minimize sigmoid loss with ADAM optimizer with learning rate 0.0001 as advised in its original paper. For all methods, we fix the weight decay param at 0.0005.

| Dataset | $(\text{TED})^n$ | BBE* | DEDPUL* | EN* | KM2 | TiCE |
|---|---|---|---|---|---|---|
| concrete | **0.071** | 0.152 | 0.176 | 0.239 | 0.099 | 0.268 |
| mushroom | **0.001** | 0.015 | 0.014 | 0.013 | 0.038 | 0.069 |
| landsat | 0.022 | 0.021 | **0.012** | 0.080 | 0.037 | 0.027 |
| pageblock | **0.007** | 0.066 | 0.041 | 0.135 | 0.008 | 0.298 |
| spambase | **0.006** | 0.047 | 0.077 | 0.127 | 0.062 | 0.276 |

Table 7: Absolute estimation error when $\alpha$ is 0.5. "*" denote oracle early stopping as defined in Sec. 6. Results reported by aggregating absolute error over 10 epochs.

| Dataset | $(\text{TED})^n$ (unknown $\alpha$) | CVuO (known $\alpha$) | PvU* (known $\alpha$) | DEDPUL* (unknown $\alpha$) | nnPU (known $\alpha$) | uPU* (known $\alpha$) |
|---|---|---|---|---|---|---|
| concrete | **86.3** | 80.1 | 83.1 | 83.7 | 83.2 | 84.4 |
| mushroom | 96.4 | 96.3 | **98.7** | **98.7** | 97.5 | 93.9 |
| landsat | **93.8** | 93.1 | 93.4 | 92.4 | 92.9 | 92.3 |
| pageblock | **95.7** | **95.7** | 95.1 | 94.5 | 93.9 | 93.9 |
| spambase | **89.4** | 88.1 | 89.2 | 86.8 | 88.5 | 87.7 |

Table 8: Accuracy for PvN classification with PU learning. "*" denote oracle early stopping as defined in Sec. 6. Results reported by aggregating aggregating over 10 epochs.

On 4 out of 5 UCI datasets, our proposed methods are better than the best performing alternatives (Table 7 and Table 8).

## G.8 Experiments on MNIST Overlap

Similar to binarized MNIST, we create a new dataset called MNIST Overlap, where the positive class contains digits from 0 to 7 and the negative class contains digits from 3 to 9. This creates a dataset with overlap between positive and negative support. Note that while the supports overlap, we sample images from the overlap classes with replacement, and hence, in absence of duplicates in the dataset, exact same images don't appear both in positive and negative subsets.

We train MLP with the same hyperparameters as before. Our findings in Table 9 and Table 10 highlight superior performance of the proposed approaches in the cases of support overlap.

| Dataset | $(TED)^n$ | BBE* | DEDPUL* | EN* | KM2 | TiCE |
|---------|-----------|------|---------|-----|-----|------|
| MNIST Overlap | **0.035** | 0.100 | 0.104 | 0.196 | 0.099 | 0.074 |

Table 9: Absolute estimation error when $\alpha$ is 0.5. "*" denote oracle early stopping as defined in Sec. 6. Results reported by aggregating absolute error over 10 epochs.

| Dataset | $(TED)^n$ (unknown $\alpha$) | CVuO (known $\alpha$) | PvU* (known $\alpha$) | DEDPUL* (unknown $\alpha$) | nnPU (known $\alpha$) | uPU* (known $\alpha$) |
|---------|-----------|------|------|---------|------|------|
| MNIST Overlap | **79.0** | 78.4 | 77.4 | 77.5 | 78.6 | 78.8 |

Table 10: Accuracy for PvN classification with PU learning. "*" denote oracle early stopping as defined in Sec. 6. Results reported by aggregating aggregating over 10 epochs.