# OpenReview forum: "Mixture Proportion Estimation and PU Learning:A Modern Approach"
_NeurIPS.cc/2021/Conference — NeurIPS 2021 Spotlight_

### Official Review · Reviewer_31y4 · 2021-07-02

**Rating:** 6
**Confidence:** 3

**Summary:**

This paper targets the problems of mixture proportion estimation and PU learning. The authors propose two simple but effective methods to address two problems. Also, the combination of two methods is proposed to train an accurate binary classifier under mild conditions. A series of experiments are conducted to verify the effectiveness of the proposed method.

**Limitations And Societal Impact:**

The **limitations and concerns** in my view are as follows.
- The authors argue the proposed method only relies on milder conditions. Such a description is somewhat not intuitive. A explanation at a high level is encouraged and expected.
- More detailed analyses for experimental results are suggested.
- The procedure of the proposed method is relatively simple. However, it still is comprised of some parts. Would the method bring a lot of extra computational consumption?
- The method consists of some hyper-parameters needed to be determined. The analyses for them (e.g. an ablation study) should be added to discuss their influence in experiments.


**Main Review:**

**Contributions and Novelty**

This paper proposes two advanced methods to tackle the problem of mixture proportion estimation and PU learning, i.e., BBE and CVuO. BBE produces consistent estimates $\hat{\alpha}$ under mild assumptions and achieves a $\mathcal{O}(1/\sqrt{n})$ convergence rate under some assumptions. Moreover, CVuO discards a certain proportion of training examples during training, which reduces the overfitting to the unlabeled positive ones. Finally, the method TED that combines the BBE and CVuO is presented to train a binary classifier under the settings of PU learning. The proposed methods are effective and somewhat novel.

**Quality**
- Theory. Detailed theoretical analyses are provided for the method BBE, which shows that, with high probability, the estimation is close to the ground truth.
- Experiments. The experimental results are convincing. The proposed method outperforms baselines in most cases.

**Clarity**

- Motivation. Although the proposed method works well, the motivation of this paper needs a clearer explanation.
- Comparison with related works. The authors state that the issues of the prior methods, e.g., complexity or strong assumptions, and aims to address these issues. However, a clearer explanation for their issues is expected to present. At the present stage (Section 2), it is still a bit hard to understand the weaknesses of the prior methods. The descriptions of experimental settings are appreciated.

**Time Spent Reviewing:**

8

---

> ### Author Response · Authors · 2021-08-10
> **Response to Reviewer 31y4**
>
> Thanks for your detailed review and positive assessment.
>
> **“More detailed analyses for experimental results are suggested. The method consists of some hyper-parameters needed to be determined. The analyses for them (e.g. an ablation study) should be added to discuss their influence in experiments.”**
>
> To analyze our results and proposed algorithms, we perform the following experiments:
>
> (i) Results in main paper with 5 different datasets with alpha = 0.5;
> (ii) Ablations with varying the mixture proportion alpha in Appendix F.4 (Figure 10);
> (iii) Ablations with varying warm start iterations W in Appendix F.4 (Figure 9);
> (iv) Results on MNIST with simulated overlaps in Appendix F.6. We create a new dataset called MNIST Overlap, where the positive class contains digits from 0 to 7 and the negative class contains digits from 3 to 9. This creates a dataset with an overlap between positive and negative support;
> (v) Results with UCI datasets in Appendix F.5.
>
>
> * Ablations with varying W, show that our procedure is not sensitive to warm start iterations and in many tasks with W = 0, we observe minor-to-no differences in the performance of (TED)^n (Lines 300-301).
>
> * Ablations with varying alpha show that our method (TED)^n maintains superior performance as compared to alternate algorithms.
>
> * Experiments on Overlap MNIST and UCI datasets show that our algorithm continues to obtain superior performance as compared to alternate algorithms.
>
> We would be happy to run additional experiments if the reviewer has any suggestions.
>
>
> **”The procedure of the proposed method is relatively simple. However, it still is comprised of some parts. Would the method bring a lot of extra computational consumption?”**
>
> In our (TED)^n procedure, we alternate between mixture proportion estimation and then using the updated estimate to train the classifier for one epoch.
>
> Because in each round of (TED)^n we update our mixture proportion estimate with BBE and then fine-tune our classifier under the CVuO objective for one epoch (using the new estimate), the computational cost is roughly the same as under ordinary training.  Breaking it down, the cost of BBE is primarily the cost of inference on validation data and the computational cost of updating the classifier with CVuO is roughly the same as training an ordinary classifier for one epoch. Thus, the computational costs associated with (TED)^n are similar to those of training and validating a standard positive versus negative classifier for $n$ epochs.
>
>
> **”The authors argue the proposed method only relies on milder conditions. Such a description is somewhat not intuitive. An explanation at a high level is encouraged and expected.”**
>
> To guarantee consistency, existing methods that leverage a blackbox classifier discuss Bayes optimality of the PvU classifier as a sufficient condition. However, we show that even in a simple toy setup, PvU training doesn’t recover the Bayes optimal classifier  (Lines 196-202).
>
> On the other hand, we need the classifier to satisfy the assumption that there exists a top bin that mostly contains positive examples—irreducible bias grows proportional to the noise in the top bin, and 0 noise is required in order for BBE to obtain consistent estimates (Lines187-191). We provide empirical evidence highlighting that deep learning classifiers indeed produce top bins that mostly contain positive examples (Fig 2(a), Lines 192-195).
>
>
> **”Although the proposed method works well, the motivation of this paper needs a clearer explanation. Section 2, it is still a bit hard to understand the weaknesses of the prior methods"**
>
> * For MPE, classical methods break down in high-dimensional settings, while recent proposals either lack theoretical coherence or depend precariously on tuning hyperparameters that are, by the very problem setting, untunable (Lines 39-48). This motivated our BBE algorithm.
>
> * For PU learning, we observe that existing methods leave a substantial accuracy gap when compared to a model trained just on the positive and negative (from the unlabeled) data in App. F.1 (Lines 218-220). To decrease this gap, we propose our CVuO algorithm.
>
> As per your suggestion, we will improve the exposition on our motivation and the issues with prior methods in the final draft.

---

> > ### Comment · Reviewer_31y4 · 2021-08-16
> > **Thanks for the response**
> >
> > Dear authors:
> >
> > Thanks for your detailed and careful reply to my review comments. I have also read the other reviews for this paper and still stand by this paper.
> >
> > Best,
> > Reviewer 31y4

---

### Official Review · Reviewer_HMxc · 2021-07-13

**Rating:** 8
**Confidence:** 3

**Summary:**

The paper proposes two methods for PU learning. First, a method for estimating the fraction of positives among unlabeled examples, named Best Bin Estimation (BBE). Secondly, an approach for PU learning called Conditional Value Under Optimism (CVuO). The BBE works on top of the decision score of a given pretrained classifier. The score is used to find a region of the instance space with the minimal ratio of the portion of unlabeled and the portion of positive examples. The ratio is an upper bound on the fraction of positives being estimated. In case there exists a region populated only by positive examples (the condition called pure positive bin property), the bound is tight. The authors provide a finite sample bound for their estimator. The CVuO turns any supervised method to PU learning algorithm. The approach assumes the fraction of positive examples to be known. The idea is to rank the unlabeled examples by a score of a pretrained classifier and remove it the fraction of positive examples. The remaining examples are taken as true negatives and then used with the true positives to re-train the classifier. The procedure is repeated until convergence. Besides, the authors propose an algorithm, called $TED^n$, combining BBE and CVuO. The methods are empirically shown to outperform several recent methods on semisynthetic problems created from CIFAR, MNIST, and IMDB datasets.

**Limitations And Societal Impact:**

I don't see any potentially negative societal impact of the work.

**Main Review:**

Originality. The idea of finding the "pure positive bin" and its usage for the estimation of the fraction of positives is to my knowledge novel. The CVuO is a straightforward heuristic to convert PU learning into supervised learning, however, I am not aware of its appearance in the literature.

Quality. The paper is technically sound.

Clarity. I am missing a high level explanation of the core idea behind the BBE. The reader is in general left on its own to extract the ideas from a dense text. On the other hand, I understand it is not easy given the limited space. I would suggest to start with showing the equation $q_u(z)=\alpha q_p(z) + (1-\alpha) q_n(z)$ which is easy to understand, and one can immediately see form it that $q_u(z)/q_p(z)\geq \alpha$ and that the bound is tight if $q_n(z) = 0$. The function of the hyper-parameter $\gamma$ of Algorithm 1 is unclear. I do not fully see the logic behind the name "Conditional Value Under Optimism". I had also an impression that the limits of the proposed method are not emphasized, namely, that it requires a prediction model nearly perfectly separating the classes for the top ranked instances.

Significance. The proposed methods are simple and on standard benchmarks outperform several recent approaches. On the other hand, the methods are tested only on data favourable for the proposed method, i.e., almost perfectly separable classes and the fraction of positives equal to 0.5.

Experiments. As I already mentioned, the methods are tested in the simplest setting when the fraction of positives is just 0.5. It is known that PU methods tend to fail when the classes are imbalanced and it is important to know how the proposed method behaves. The errors are reported without any measure of uncertainty and hence it is hard to judge significance of the the observed differences. The evaluation protocol is not completely clear, but the details are perhaps in the appendix.

Typos:
- line 30: "of of"
- line 114: ". in our experiments"
- Algo 3, line 4: the same iterator "i" is used in both loops
- Algo 3, line 14: "Lines 4-7" -> "Lines 4-9"

**Time Spent Reviewing:**

3-4

---

> ### Author Response · Authors · 2021-08-10
> **Response to Reviewer HMxc**
>
> Thank you for your positive assessment and constructive feedback on our work.
>
> **”...  the methods are tested only on data almost perfectly separable classes and the fraction of positives equal to 0.5.”**
>
> In the Appendix, we have included additional results. In particular, we include:
> (i) Results with varying the mixture proportion $\alpha$ in Appendix F.4 (Figure 10);
> (ii) Results on MNIST with simulated overlaps in Appendix F.6. We create a new dataset called MNIST Overlap, where the positive class contains digits from 0 to 7 and the negative class contains digits from 3 to 9. This creates a dataset with an overlap between positive and negative support;
> (iii) Results with UCI datasets in Appendix F.5
>
> As per your suggestion, we will add a summary of the results from these ablations in the main paper in the final version.
>
>
> **”I am missing a high-level explanation of the core idea behind the BBE.”**
>
> We apologize for the confusion. Based on your suggestions, we will elaborate on the core idea behind BBE to improve the exposition in the final version.
>
>
> **”The function of the hyper-parameter of Algorithm 1 is unclear”**
>
> We apologize for the confusion. We use $\gamma$ to get tight theoretic guarantees (Appendix B.1). In Algo 1, we add the confidence bound multiplied by $(1/\widehat{q}_p(\widehat {c}) - 1/\gamma)+$ to our ratio estimate $ \widehat{q}_u(\widehat {c}) /  \widehat{q}_p(\widehat {c})$ to yield an estimate with tighter guarantees, i.e.,  when $\widehat{q}_p(\widehat {c})$ is smaller than $1/\gamma$, adding upper confidence bound scaled by the multiplier $(1/\widehat{q}_p(\widehat {c}) - 1/\gamma )$ avoids $1/\widehat{q}_p(\widehat {c})$ in our upper bound in Theorem 1 (Lines 490-493).
>
> However, as mentioned in Appendix B.1 (Lines 494-495), since we are minimizing the upper confidence bound (in Line 3 in Algo 1), we never observe $\widehat{q}_p(\widehat {c})$ taking small values than $\gamma = 0.1$ (fixed throughout all of our experiments). Hence, the gamma term has no effect in experiments.
>
> Motivated by this finding, we have recently improved the analysis to derive a lower bound on $\widehat q_p(\widehat c)$. We show that $\widehat {q}_p(\widehat{c}) = \Omega  (\widehat{q}_p (c^*))$ and thereby we drop the upper confidence bound term (i.e., second term in the estimate $\widehat{\alpha}$ in Algo 1 and the new estimate is given by $\widehat{\alpha} = \widehat{q}_u(\widehat {c}) /  \widehat{q}_p(\widehat {c})$).
>
>
> **”... logic behind the name "Conditional Value under Optimism is unclear"**
>
> The name is motivated by Conditional Value at Risk (CVaR). Similar to CVaR, we discard samples from the tail of the distribution under the optimistic assumption that samples with highest loss (i.e. in the tail) are positives.
>
>
> **”... limits of the proposed method are not emphasized, namely, that it requires a prediction model nearly perfectly separating the classes for the top-ranked instances”**
>
> Yes, we need the classifier to satisfy the assumption that there exists a top bin that mostly contains positive examples—irreducible bias grows proportional to the fraction of negatives in the top bin (Theorem 1), and 0 noise is required in order for BBE to obtain consistent estimates (Lines187-191). We provide empirical evidence highlighting that deep learning classifiers indeed produce top bins that mostly contain positive examples (Fig 2(a), Lines 192-195).
>
> We will make the limitation of BBE explicit in the final version.
>
> **”The errors are reported without any measure of uncertainty and hence it is hard to judge the significance of the observed differences.”**
>
> Thanks for pointing out this oversight. We have updated the draft with measures of uncertainty over multiple runs (with 3 seeds) and present the results from the updated Tables 1 and 2 with measures of uncertainty below:
>
> | Dataset          | Model  | (TED)^n            | BBE*             | Dedpul*          | KM2   | TiCE  |
> |------------------|--------|--------------------|------------------|------------------|-------|-------|
> | Binarized CIFAR  | ResNet | 0.018 $\pm$ 0.0025 | 0.072$\pm$  0.0006 | 0.075 $\pm$ 0.0023 | 0.181 | 0.251 |
> | CIFAR Dog vs Cat | ResNet | 0.074 $\pm$ 0.014    | 0.12 $\pm$ 0.009   | 0.113 $\pm$ 0.012  | 0.11  | 0.203 |
> | Binarized MNIST  | FCN    | 0.021 $\pm$ 0.003    | 0.028 $\pm$ 0.002  | 0.027 $\pm$ 0.001  | 0.102 | 0.247 |
> | MNIST 17         | FCN    | 0.003 $\pm$ 0.001    | 0.008 $\pm$ 0.004  | 0.006 $\pm$ 0.002  | 0.065 | 0.117 |
> | IMDb             | BERT   | 0.008 $\pm$ 0.002    | 0.011 $\pm$ 0.003  | 0.016 $\pm$ 0.004  | -     | -     |
>
>
> | Dataset          | Model  | (TED)^n         | CVuO          | PvU*          | Dedpul*       | nnPU          |
> |------------------|--------|-----------------|---------------|---------------|---------------|---------------|
> | Binarized CIFAR  | ResNet | 82.7 $\pm$ 0.20 | 82.6 $\pm$ 0.44 | 78.3 $\pm$ 0.71 | 78.4 $\pm$ 0.54 | 76.8 $\pm$ 0.73 |
> | CIFAR Dog vs Cat | ResNet | 76.1 $\pm$ 0.42 | 74.0 $\pm$ 0.61 | 71.6 $\pm$ 1.43 | 70.9 $\pm$ 1.69 | 72.6 $\pm$ 0.82 |
> | Binarized MNIST  | FCN    | 95.9 $\pm$ 0.1  | 96.4 $\pm$ 0.11 | 94.5 $\pm$ 0.24 | 95.2 $\pm$ 0.14 | 95.9 $\pm$ 0.16 |
> | MNIST 17         | FCN    | 98.6 $\pm$ 0.11 | 98.6 $\pm$ 0.08 | 93.7 $\pm$ 2.04 | 98.1 $\pm$ 0.01 | 98.2 $\pm$ 0.18 |
> | IMDb             | BERT   | 87.6 $\pm$ 0.37 | 87.4 $\pm$ 0.52 | 86.1 $\pm$ 0.63 | 87.3 $\pm$ 0.19 | 86.2 $\pm$ 0.15 |
>
>
> **”The evaluation protocol is not completely clear, but the details are perhaps in the appendix.”**
>
> Yes, we include additional experimental details in Appendix E. As per your suggestions, we will include a more detailed summary of the evaluation protocol in the main paper in the final draft.
>
> Thanks for catching the typos. We have fixed them in our draft.

---

> > ### Comment · Reviewer_HMxc · 2021-08-16
> > **Thanks for the reply**
> >
> > The authors have replied all my concerns raised in the review in very detailed manner. I believe the paper should be accepted.

---

### Official Review · Reviewer_DmAg · 2021-07-16

**Rating:** 8
**Confidence:** 4

**Summary:**

This paper proposes a unified framework for learning from positive-unlabeled (PU) data, including BBE for mixture proportion estimation (MPE)  and CVuO for PU learning. In the literature there are important and elegant works exists that analyze the consistency of the mixture proportion without the irreducibility assumption. They formalize their analyses based on an assumption that the percentage of a blackbox classifier's output for positive samples greater than a fixed value is higher than that for negative samples. I suggest a brief introduction of the conditions that this assumption satisfies, maybe the inductive bias or something, while intuitively such an assumption is quite reasonable. They provide finite sample guarantee for BBE and also empirically verify it. In the PU learning stage, the authors use an iterative manner to progressively identify and remove all the positive points from the unlabeled data during training. This is again quite a reasonable way of thinking because of the memorization of neural networks. The combination of BBE and CVuO is called (TED)^n. The authors provide a lot of supportive experiments to verify their methods.

**Limitations And Societal Impact:**

Yes.

**Main Review:**


Two main tasks have been solved in this paper. The first relaxes the strong irreducibility assumption in MPE. Since we don't have any prior information about Pn, it is hard to be verified whether the assumption is satisfied in a specific problem. If not, the existing MPE methods may suffer from estimation bias. BBE is just a clever way of solving that. The second uses a simple manner establishing the state of the art. Overall, new theoretical results provide important insights on an important topic. The paper is clear and well written. There is a good balance of theoretical findings and empirical validations.

Minor suggestions:
- In Algorithm 1, \hat{\alpha} cause some doubts. How to intuitively get this equation? From the supplementary materials, it seems to be a derivation back from the theoretical result? Besides, it would be better to introduce the notions that appear for the first time, e.g., ()_{+}.
- In (TED)^n, how to make a reasonable division between the training set and hold-out set? Should they be evenly split?

**Time Spent Reviewing:**

7

---

> ### Author Response · Authors · 2021-08-10
> **Response to Reviewer DmAg**
>
> Thanks for your positive feedback and for championing our paper. We respond to your specific concerns below:
>
> **“... \hat{\alpha} cause some doubts. How to intuitively get this equation? From the supplementary materials, it seems to be a derivation back from the theoretical result?”**
>
> We apologize for the confusion here. Yes, the derivation for $\widehat{\alpha}$ in Algo 1 is motivated by our theoretical analysis. We add the confidence bound to our ratio estimate $ \widehat{q}_u(\widehat {c}) /  \widehat{q}_p(\widehat {c})$ to yield an estimate with tighter guarantees, i.e.,  when $\widehat{q}_p(\widehat {c})$ is smaller than $1/\gamma$, adding upper confidence bound scaled by the multiplier $(1/\widehat{q}_p(\widehat {c}) - 1/\gamma )$ avoids $1/\widehat{q}_p(\widehat {c})$ (which, in principle, can be arbitrarily small) in our upper bound in Theorem 1 (Lines 490-493).
>
> However, as mentioned in Sec B.1 in Appendix (Lines 494-495), since we are minimizing the upper confidence bound (in Line 3 in Algo 1), we never observe $\widehat{q}_p(\widehat {c})$ taking small values than $\gamma = 0.1$ (fixed throughout our experiments). Hence, the second term with gamma has no effect in experiments.
>
> Motivated by this finding, we have recently improved the analysis to derive a lower bound on $\widehat q_p(\widehat c)$. We show that $\widehat {q}_p(\widehat{c}) = \Omega  (\widehat{q}_p (c^*))$ and thereby we drop the upper confidence bound term (i.e., second term in the estimate $\widehat{\alpha}$ in Algo 1 and the new estimate is given by $\widehat{\alpha} = \widehat{q}_u(\widehat {c}) /  \widehat{q}_p(\widehat {c})$).
>
>
> **“In (TED)^n, how to make a reasonable division between the training set and hold-out set? Should they be evenly split?”**
>
> Since the training set is used to learn the classifier (parameters of a deep neural network) and the hold-out set is just used to learn the mixture proportion estimate $\widehat{\alpha}$ (scalar), we use a larger dataset for training. Throughout the experiments, we use an 80-20 split of the original set. These are important details and we will be sure to include them in the final version.
>
> At a high level, we have an error bound on the mixture proportion estimate and we can use that to decide the split. As long as we use enough samples to make the $\mathcal{O}(1/\sqrt{n})$ small, we can use the rest of the samples to learn the classifier.

---

> > ### Comment · Reviewer_DmAg · 2021-08-16
> > **Response to authors**
> >
> > Thank you. I very much appreciate the detailed response.

---

### Official Review · Reviewer_sRZG · 2021-07-18

**Rating:** 7
**Confidence:** 4

**Summary:**

This paper tackles the well known problem of mixture proportion estimation and Positive-Unlabelled (PU) learning. The authors propose a a (new) estimator for MPE based on the assumption of a PU classifier having a "pure" top bin. The authors then develop a novel  iterative outlier-removal-like algorithm for learning a classifier separating the positives and negatives from PU data assuming access to the true MPE. Finally, they combine these two approaches to create a novel alternating algorithm that estimates MPE based on a current PvN classifier, and use the MPE estimate to learn a better PvN classifier from PU data.

**Limitations And Societal Impact:**

No known negative social impact is conceivable for this work specifically, as it is primarily algorithmic in nature for solving a well-known problem.

**Main Review:**

The CvUO algorithm and TED procedure are novel, interesting, and deserve credit.

The BBE algorithm, on the other hand, seems extremely similar to other approaches, e.g. Scott (2015) gives an approach of learning a PvU classifier, and using held-out data to estimate the ROC (with confidence intervals) and give the smallest slope of the ROC curve to (1,1) as an MPE estimate.  This is, essentially, the same as finding a threshold and taking the (confidence adjusted) ratio of q_p/q_u. There are some subtle differences, but they seem rather negligible. While Scott (2015) uses Kernel logistic regression to generate the ROC, there is nothing preventing the use of deep learning methods to generate the ROC curve. The assumptions of irreducibility, i.e. existence of a set S such that P_n(S)=0 but P_p(S)>0 is exactly the same as the existence of a pure top bin when the sets S are generated by thresholding a learned PvU classifier.

The CvUO algorithm seems to essentially assume equal class priors between positive and negative. For example, if alpha=0, CvUO should just return a standard PvN classifier, but the loss function used weights the total positive samples as equal to total negative samples regardless of the actual number of samples of each type. A similar issue exists with the proof of proposition 1 in appendix D. I am guessing this is a bug and can be fixed.

Some guarantee of the sort to ensure that CvUO algorithm does not get progressively worse would be valuable. For example if in one iteration a majority of the alpha fraction of sample removed are not negative the classifier learned after that epoch might be even worse than the previous epoch and start a negative spiral. Can a guarantee be given for this to not happen? The proof in Appendix D does NOT take care of this.





**Time Spent Reviewing:**

5

---

> ### Author Response · Authors · 2021-08-10
> **Response to Reviewer sRZG**
>
> Thank you for your positive assessment and thoughtful review of our work.
>
> **”The BBE algorithm, on the other hand, seems extremely similar to other approaches, e.g. Scott (2015)”**
>
> Thanks for pointing out this connection. While there are some key similarities between our approaches, there are also some important distinctions, and our final draft will benefit from a more in-depth discussion of these points.
>
> To summarize, all of Scott (2015)’s theoretical results pertain to an estimator due to Blanchard 2010 that relies on VC bounds that are known to be loose. This method and the theoretical analysis is very different from ours. However, due to the intractability of Blanchard (2010)’s  estimator, Scott (2015) implement a second **heuristic estimator** (Scott 2015, Section 5) based on identifying a point on the AUC curve such that the slope of the line segment between this point and (1,1) is minimized. While on the surface, this approach is similar to our best bin estimator, there are some striking differences:
>
> 1. While they provide no theoretical guarantees for their **heuristic estimator**, we provide guarantees that our estimator will converge to the best estimate achievable over all choices of the bin size and provide consistent estimates whenever a pure top bin exists.
> 2. Second, while both estimates involve thresholds, the functional form of the estimates are different: Scott 2015’s estimator is the ratio of quantities obtained by binomial tail inversion (i.e. upper bound in the numerator and lower bound in the denominator) which can be biased to overestimate MPE. By contrast, the final BBE estimate is simply the ratio of empirical CDFs at the optimal threshold.
> 3. We implemented the proposal of Scott (2015) and identified that the choices in BBE
> create substantial differences in the empirical performance as shown in the table below (setup is the same as in Table 1):
>
> | Dataset          | Model  | (TED)^n | BBE   | Dedpul | Scott (2015)$^1$ |
> |------------------|--------|---------|-------|--------|--------------------------|
> | Binarized CIFAR  | ResNet | 0.018   | 0.072 | 0.075  | 0.091                    |
> | CIFAR Dog vs Cat | ResNet | 0.074   | 0.12  | 0.113  | 0.158                    |
> | Binarized MNIST  | FCN    | 0.021   | 0.028 | 0.027  | 0.063                    |
> | MNIST 17         | FCN    | 0.003   | 0.008 | 0.006  | 0.037                    |
>
> $^1$as mentioned in the Scott (2015) implementation (https://web.eecs.umich.edu/~cscott/code/mpe_v2.zip), we use the binomial inversion at \delta instead of \delta/n (rescaling using the union bound). Since we are using Binomial inversion at n discrete points simultaneously, we should use the union-bound penalty. However, using union bound penalty substantially increases the bias in their estimator.
>
> These are important points to clarify and we will add a detailed comparison of the two methods in the camera-ready version.
>
> **“The CvUO algorithm seems to essentially assume equal class priors between positive and negative”**
>
> Thanks for bringing up this important point. Indeed, we care about distinguishing between positive versus negative examples among the unlabeled set and these points are (1) not necessarily class balanced and (2) may have a different class balance from the amounts of available positive versus unlabeled data. We thought about this concern early on and in our initial experiments, attempted to address the matter via importance-weighted risk minimization (reweighting the positive loss and negative loss terms with $\widehat{\alpha}$ and $1- \widehat{\alpha}$ respectively). However, we observed no effect of multiplying the estimated mixture proportion estimate on final classification performance. Hence, similar to earlier works (Kiryo et al. 2017, Du Plessis et al, 2015), we followed equal weighting of the positive and negative loss in the final objective.
>
> We note that for deep neural networks (for which model misspecification is seldom a prominent concern) and when the underlying classes are separable (as with most image datasets), it is known that importance weighting has little to no effect on the final classifier (Byrd 2019). This may explain why IW-ERM does not confer benefits in our experiments. For completeness, we will add relevant discussion and experiments in the final draft.
>
>
> **”Some guarantee of the sort to ensure that CvUO algorithm does not get progressively worse would be valuable”**
>
> This is a very interesting suggestion. And based on your comment we thought about it and we have one result in the population case. While it is hard to argue about the one-step loss with optimization algorithms in a finite-sample case, in the population case we show that one step of our alternating procedure cannot increase the loss.
>
> Consider the following objective function,
> $$ L(f_t, w_t) = E_{x \sim P_p}[ l( f_t(x), 0) ] + E_{x \sim P_u}[ w_t(x) l( f_t(x), 1) ]  $$
> $$ \text{s.t.} E_{x \sim P_u}[ w(x)] = 1-\alpha \text{ and } w(x) \in \{0,1\}$$
>
> Given $f_t$ and $w_t$, CVuO can be summarized as the following two step iterative procedure: (i) Fix $f_t$, optimize the loss to obtain w_{t+1}; and (ii) Fix $w_{t+1}$ and optimize the loss to obtain $f_{t+1}$. By construction of CVuO, we select $w_{t+1}$ such that we discard points with highest loss, and hence $L(f_t, w_{t+1}) \le L(f_t, w_{t})$. Fixing $w_{t+1}$, we minimize the $L(f_t, w_{t+1})$ to obtain $f_{t+1}$ and hence $L(f_{t+1}, w_{t+1}) \le L(f_t, w_{t+1})$. Combining these two steps, we get $L(f_{t+1}, w_{t+1}) \le L(f_t, w_{t})$.
>
> We will include this result in the final version.
>
> [1] G. Blanchard, G. Lee, and C. Scott. Semi-supervised novelty detection. The Journal of Machine Learning Research, 2010
>
> [2] C. Scott. A rate of convergence for mixture proportion estimation, with application to learning from noisy labels. In Artificial Intelligence and Statistics, 2015.
>
> [3] M. Du Plessis, G. Niu, and M. Sugiyama. Convex formulation for learning from positive and unlabeled data. In International conference on machine learning, 2015.
>
> [4] R. Kiryo, G. Niu, M. C. Du Plessis, and M. Sugiyama. Positive-unlabeled learning with a non-negative risk estimator. In Advances in neural information processing systems, 2017
>
> [5] J. Byrd, Z. Lipton. What is the Effect of Importance Weighting in Deep Learning? In International Conference on Machine Learning, 2019

---

> > ### Comment · Reviewer_sRZG · 2021-08-15
> > **Thanks for the detailed reply**
> >
> > Thanks for the reply.
> >
> > If text on detailed comparison (including formula/algebra) of BBE with the ROC slope is added, and also a clarification on why the positive and negative priors are assumed to be equal is added, this paper would go up one level in score.

---

### Decision · Program_Chairs · 2021-09-27

**Decision:**

Accept (Spotlight)

**Comment:**


This paper proposes new methods for the related problems of mixture proportion estimation and positive-unlabeled learning, with theoretical support, and shows state of the art performance, especially for large scale problems. I tend to agree with one of the reviewers that the MPE method is not really that novel, sharing many conceptual similarities with previous ROC based methods. This should be clearly addressed in the final revision, as should all reviewer comments. In addition it would be desired to have some theory for the iterative scheme. Without such, the authors also need to address possible failure cases and limitations of $(TED)^n$. Nonetheless, there is still sufficiently novelty and merit to warrant publication.

Additional comments:

While the experimental contributions are clear, I'd like to ask the authors to comment on how the MPE theory compares to prior work. Is this theory merely "supporting", or does it offer advances in MPE theory in any substantive way?

Relevant reference: Henry Reeve and Ata Kaban. Exploiting geometric structure in mixture proportion estimation with generalised Blanchard-Lee-Scott estimators, ALT 2019